# Critical roles for 'housekeeping' nucleases in type III CRISPR-Cas immunity

**Lucy Chou-Zheng, Asma Hatoum-Aslan***

Microbiology Department, University of Illinois Urbana-Champaign, Urbana, United States

**Abstract** CRISPR-Cas systems are a family of adaptive immune systems that use small CRISPR RNAs (crRNAs) and CRISPR-associated (Cas) nucleases to protect prokaryotes from invading plasmids and viruses (i.e., phages). Type III systems launch a multilayered immune response that relies upon both Cas and non-Cas cellular nucleases, and although the functions of Cas components have been well described, the identities and roles of non-Cas participants remain poorly understood. Previously, we showed that the type III-A CRISPR-Cas system in *Staphylococcus epidermidis* employs two degradosome-associated nucleases, PNPase and RNase J2, to promote crRNA maturation and eliminate invading nucleic acids (Chou-Zheng and Hatoum-Aslan, 2019). Here, we identify RNase R as a third 'housekeeping' nuclease critical for immunity. We show that RNase R works in concert with PNPase to complete crRNA maturation and identify specific interactions with Csm5, a member of the type III effector complex, which facilitate nuclease recruitment/stimulation. Furthermore, we demonstrate that RNase R and PNPase are required to maintain robust anti-plasmid immunity, particularly when targeted transcripts are sparse. Altogether, our findings expand the known repertoire of accessory nucleases required for type III immunity and highlight the remarkable capacity of these systems to interface with diverse cellular pathways to ensure successful defense.

## Editor's evaluation

CRISPR-Cas systems are essential components of an adaptive immune system that protects bacteria and archaea from infection of foreign genetic elements like phages and plasmids. The work presented here demonstrates that some CRISPR systems (i.e., type III-A) rely on host nucleases (i.e., RNase R and PNPase) for faithful processing of CRISPR RNAs into short mature CRISPR RNA (crRNAs) that are required for defense. Collectively, this work expands our fundamental understanding of degradosome-associated nucleases, and their contribution to the adaptive immune response in bacteria.

**\*For correspondence:**
ahatoum@illinois.edu

**Competing interest:** The authors declare that no competing interests exist.

## Introduction

CRISPR-Cas (Clustered regularly interspaced short palindromic repeats-CRISPR associated) systems are adaptive immune systems in prokaryotes that use small CRISPR RNAs (crRNAs) in complex with Cas nucleases to sense and degrade foreign nucleic acids (*Barrangou et al., 2007*; *Jansen et al., 2002*; *Hille et al., 2018*). The CRISPR-Cas pathway generally occurs in three stages—adaptation, crRNA biogenesis, and interference. During adaptation, Cas nucleases clip out short sequences (known as 'protospacers') from invading nucleic acids and integrate them into the CRISPR locus as 'spacers' in between short DNA repeats. During crRNA biogenesis, the repeat-spacer array is transcribed as a long precursor crRNA (pre-crRNA), which is subsequently processed within repeats to generate mature crRNAs that each bear a single spacer sequence. Mature crRNAs combine with one or more Cas nucleases to form effector complexes, which, during interference, detect and cleave

matching nucleic acid invaders. Although all CRISPR-Cas systems follow this general pathway, they exhibit striking diversity in the composition of their effector complexes and mechanisms of action. Accordingly, they have been divided into two classes, six types (I–VI), and over 30 subtypes (*Makarova et al., 2020b*; *Koonin and Makarova, 2022*).

Type III CRISPR-Cas systems are the most closely related to the ancestral system from which all class I systems have evolved and are arguably the most complex (*Mohanraju et al., 2016*; *Koonin and Makarova, 2022*). Type III systems typically utilize multi-subunit effector complexes that recognize foreign RNA and coordinate a sophisticated immune response that results in the destruction of the invading RNA and DNA. Of the six subtypes currently identified (A–F), types III-A and III-B are the best characterized. In these systems, crRNA binding to a complementary transcript triggers at least three catalytic activities by members of the effector complex: target RNA shredding by Cas7/Csm3/Cmr4 (*Hale et al., 2009*; *Staals et al., 2013*; *Staals et al., 2014*; *Samai et al., 2015*; *Tamulaitis et al., 2014*), nonspecific DNA degradation by Cas10 (*Samai et al., 2015*; *Kazlauskiene et al., 2016*; *Estrella et al., 2016*; *Liu et al., 2017*; *Elmore et al., 2016*), and Cas10-catalyzed production of cyclic-oligoadenylates (cOAs), second-messenger molecules that bind and stimulate accessory nucleases outside of the effector complex (*Niewoehner et al., 2017*; *Kazlauskiene et al., 2017*; *Han et al., 2018*; *Nasef et al., 2019*). Such accessory nucleases typically possess CRISPR-associated Rossman Fold (CARF) domains to which cOAs bind and are encoded within or proximal to the type III CRISPR-Cas locus (*Shmakov et al., 2018*; *Shah et al., 2019*; *Makarova et al., 2020a*). Indeed, recent studies have relied upon these two features to discover new cOA-responsive accessory nucleases and validate their contributions to type III defense (*Han et al., 2018*; *Athukoralage et al., 2019*; *McMahon et al., 2020*; *Rostøl et al., 2021*; *Zhu et al., 2021*). However, as the list of CRISPR-associated accessory nucleases continues to grow, the identities and contributions of non-Cas participants in type III immunity remain poorly understood.

Our previous work showed that the type III-A CRISPR-Cas system in *Staphylococcus epidermidis* (herein referred to as CRISPR-Cas10) employs the 'housekeeping' nucleases PNPase and RNase J2 during multiple steps in the immunity pathway (*Walker et al., 2017*; *Chou-Zheng and Hatoum-Aslan, 2019*; *Figure 1A–C*). During crRNA biogenesis, Cas6 cleaves pre-crRNAs within repeats to generate 71 nucleotide (nt) intermediates (*Hatoum-Aslan et al., 2014*), and these intermediates are processed on their 3'-ends by PNPase and one or more unidentified nuclease(s) to produce mature species of 43, 37, and 31 nt in length (*Chou-Zheng and Hatoum-Aslan, 2019*). We also showed that PNPase helps prevent phage nucleic acid accumulation during an active infection, implying a direct role for PNPase during interference. While searching for additional maturation nuclease(s), we fortuitously identified RNase J2 as another player in the pathway; however, while it has little/no effect on crRNA maturation, RNase J2 is essential for interference against phage and plasmid invaders. Notably, PNPase and RNase J2 are members of the RNA degradosome, a highly conserved complex of ribonucleases, helicases, and metabolic enzymes primarily involved in RNA processing and decay (*Tejada-Arranz et al., 2020*). Our original observation that these nucleases co-purify in trace amounts with the Cas10-Csm complex in *S. epidermidis* (*Walker et al., 2017*) led to the discovery of their additional contributions to CRISPR-Cas defense.

Here, we sought to complete the crRNA maturation pathway in *S. epidermidis* and discovered that RNase R is the second (and final) nuclease necessary for the process. We demonstrate that RNase R works in concert with PNPase to catalyze crRNA maturation in a purified system, and these enzymes work synergistically in the cell to maintain robust anti-plasmid immunity. Furthermore, we identified specific interactions between these 'housekeeping' nucleases and Csm5 (a member of the Cas10-Csm complex within the Cas7 group), which facilitate their recruitment and/or stimulation. Altogether, our findings expand the known repertoire of non-Cas nucleases that facilitate type III CRISPR-Cas defense and highlight the remarkable capacity of this system to interface with diverse nondefense cellular pathways to maintain robust immunity.

## Results

### RNase R and PNPase are necessary for crRNA maturation in the cell

Previously, we showed that an in-frame deletion of *pnp* (which encodes PNPase) in *S. epidermidis* causes loss of about half of the mature crRNA species and significant (approximately tenfold)

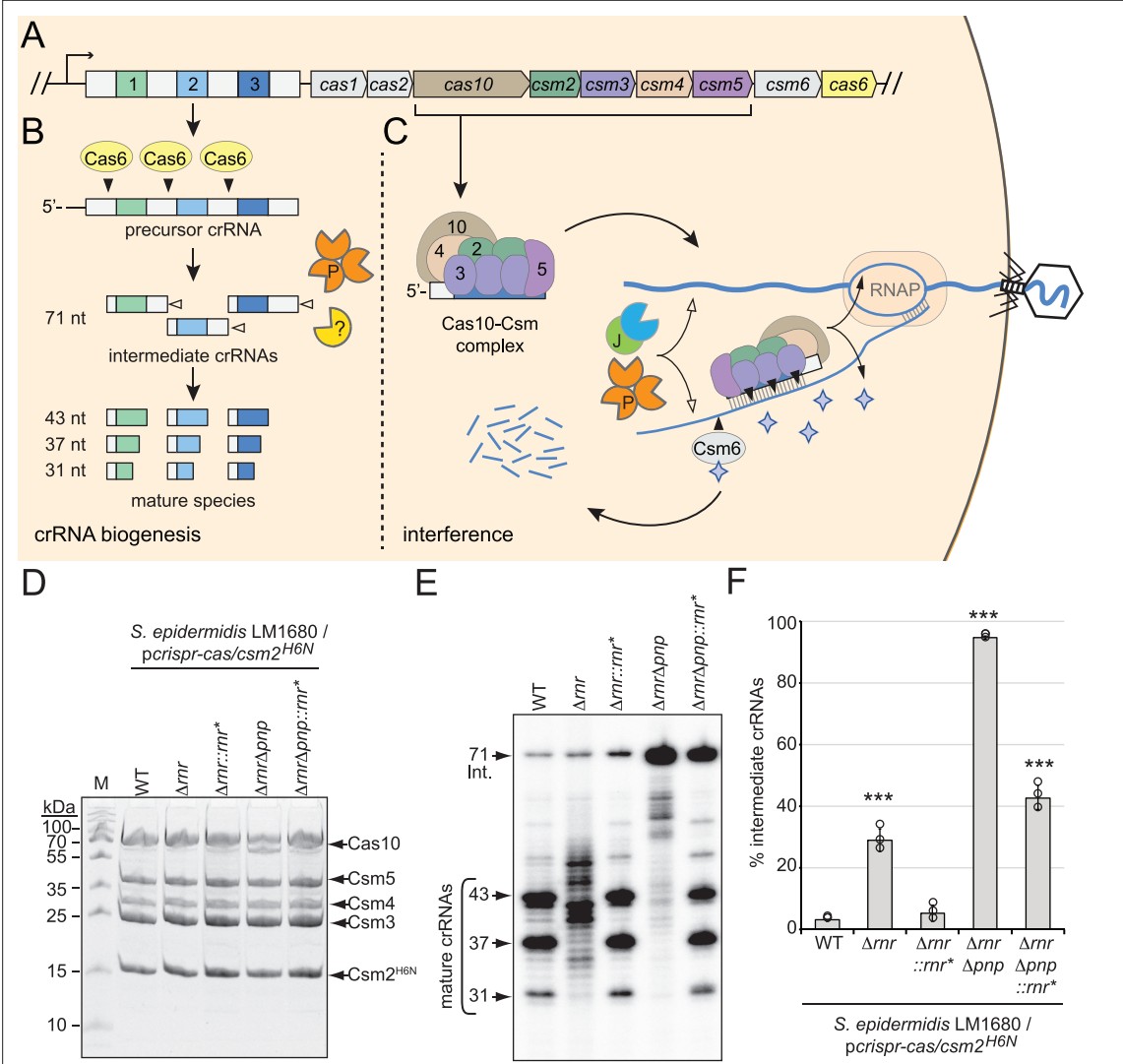

**Figure 1.** RNase R and PNPase are necessary for crRNA maturation in the cell. (**A**) The type III-A CRISPR-Cas system (herein referred to as CRISPR-Cas10) in *S. epidermidis* RP62a encodes three spacers (colored squares), four repeats (light gray squares), and nine CRISPR-associated (*cas* and *csm*) genes (colored pentagons). (**B**) During crRNA biogenesis, the repeat-spacer array is transcribed into a precursor crRNA and processed into mature species in two steps. In the first step, the endoribonuclease Cas6 cleaves within repeat sequences to generate intermediate crRNAs of 71 nt in length. In the second step, intermediates are trimmed on their 3'-ends by PNPase and other unknown nuclease(s), which are the subject of this study. These activities generate mature crRNAs that range from 43 to 31 nt in length. (**C**) Mature crRNAs associate with Cas10, Csm2, Csm3, Csm4, and Csm5 in various stoichiometries to form the Cas10-Csm effector complex. Interference is initiated when the effector complex binds to invading transcripts that bear complementarity to the crRNA. During interference, invading DNA and RNA are degraded by CRISPR-associated (Cas) and non-Cas nucleases (see text for details). Filled triangles illustrate events catalyzed by Cas enzymes, and open triangles illustrate events catalyzed by non-Cas nucleases. P, PNPase; J, RNase J1/J2; RNAP, RNA polymerase. Purple stars represent cyclic oligoadenylate molecules produced by Cas10. (**D**) Cas10-Csm complexes extracted from indicated *S. epidermidis* LM1680 strains bearing p*crispr-cas/csm2*H6N are shown. The plasmid p*crispr-cas* contains the entire CRISPR-Cas10 system with a 6-His tag on the N-terminus of Csm2. Whole-cell lysates from indicated strains were subjected to Ni2+ affinity chromatography, and purified complexes were resolved in an SDS-PAGE gel and visualized with Coomassie G-250 staining. M, denaturing protein marker; kDa, kilodalton. See *Figure 1—source data 1*. (**E**) Total crRNAs associated with Cas10-Csm complexes in panel (**D**) are shown. Complex-bound crRNAs were extracted from complexes, radiolabeled at their 5'-ends, and resolved on a denaturing gel. See *Figure 1—source data 2*. (**F**) Fractions of complex-bound intermediate crRNAs relative to total crRNAs are shown for indicated strains. The percent intermediate crRNAs represents the ratio of the intermediate (71 nt) band density to the sum of band densities of the major crRNA species (71, 43, 37, and 31 nt). Data shown represents an average of three independent trials (± S.D). A two-tailed *t*-test was performed to determine significance and *** indicates p<0.0005. See *Figure 1—source data 3*.

The online version of this article includes the following source data and figure supplement(s) for figure 1:

**Source data 1.** Raw uncropped image for panel D.

**Source data 2.** Raw uncropped image for panel E.

*Figure 1 continued on next page*

*Figure 1 continued*

**Source data 3.** Percent intermediate crRNAs for individual replicates in panel F.

**Figure supplement 1.** Confirmation of *rnr* knock-out and knock-in *S. epidermidis* strains.

**Figure supplement 1—source data 1.** Raw uncropped image for panel B.

**Figure supplement 1—source data 2.** Raw uncropped image for panel D.

accumulation of intermediates (*Chou-Zheng and Hatoum-Aslan, 2019*), indicating that one or more additional nucleases contribute to crRNA maturation. We also showed previously that PNPase co-purifies with the Cas10-Csm complex in sub-stoichiometric amounts along with at least five additional cellular nucleases that serve as maturation nuclease candidates—RNase J1, RNase J2, Cbf1, RNase R, and RNase III (*Walker et al., 2017*). Given that crRNA maturation relies upon 3′–5′ exonuclease activity, here we sought to investigate the two remaining nucleases in the list that possess this function—Cbf1 and RNase R. Unfortunately, repeated attempts to delete *cbf1* from *S. epidermidis* failed, suggesting that it may be essential for cell viability under standard laboratory growth conditions. However, *rnr* (which encodes RNase R) was readily deleted in the clinical isolate *S. epidermidis* RP62a (*Christensen et al., 1987*) as well as in *S. epidermidis* LM1680, a mutant variant of RP62a that has lost the CRISPR-Cas system (*Jiang et al., 2013*; *Figure 1—figure supplement 1A and B*). To determine the extent to which RNase R contributes to crRNA maturation in vivo, a plasmid that encodes the type III-A CRISPR-Cas system of RP62a, p*crispr-cas/csm2*$^{H6N}$ (*Hatoum-Aslan et al., 2013*) was introduced into *S. epidermidis* LM1680/Δ*rnr*. Importantly, this construct encodes a 6-histidine (6-His) tag on the N-terminus of Csm2, which allows for complex purification via Ni$^{2+}$-affinity chromatography. Cas10-Csm complexes were subsequently purified from the wild-type (WT) and Δ*rnr* strains (*Figure 1D*), crRNAs were further purified from the complexes and visualized (*Figure 1E*), and fractions of intermediate species were quantified (*Figure 1F*). This experiment revealed that deletion of RNase R alone causes complete loss of precisely processed mature species and production of crRNAs with a range of aberrant lengths. To confirm that the loss of RNase R is responsible for this phenotype, we returned *rnr* to its native locus in the genome to generate *S. epidermidis* LM1680/Δ*rnr::rnr\**—in this strain, silent mutations were introduced into *rnr* to distinguish the knock-in strain from original WT (*Figure 1—figure supplement 1C and D*). The same assays were then repeated and showed that crRNAs in the complemented strain have sizes similar to those found in WT (*Figure 1D–F*). These results demonstrate that RNase R is necessary for crRNA maturation in vivo.

In order to determine the extent to which other nuclease(s) may contribute to crRNA maturation in the absence of RNase R and PNPase, we created and tested a double-knockout. Specifically, *rnr* was deleted from the LM1680/Δ*pnp* strain (*Chou-Zheng and Hatoum-Aslan, 2019*) to generate LM1680/Δ*rnr*Δ*pnp*, and crRNAs within the complexes were examined. The results showed a complete loss of mature species in the double mutant, with ~95% of crRNAs trapped in the intermediate state (*Figure 1E and F*). In addition, when *rnr* is returned to the double mutant (i.e., in LM1680/Δ*rnr*Δ*pnp::rnr\**), some mature crRNAs are recovered with significant accumulation of the 71 nt intermediates, similar to the phenotype observed in LM1680/Δ*pnp* (*Chou-Zheng and Hatoum-Aslan, 2019*). Altogether, these data demonstrate that RNase R and PNPase are likely the primary drivers of crRNA maturation in the cell.

## RNase R and PNPase are sufficient to catalyze crRNA maturation in a purified system

Given that the deletion of RNase R on its own causes complete loss of precisely processed mature species in vivo, while deletion of PNPase still allows for some maturation to occur, the possibility exists that RNase R alone might be sufficient to catalyze maturation to completion in a purified system (i.e., in the absence of nonspecific cellular RNA substrates). Conversely, it is also possible that other nucleases in the cell (such as Cbf1) might contribute to crRNA maturation. Indeed, Cbf1 (also called YhaM) is known to work together with other 3′–5′ exonucleases in the cell to help clear RNA decay intermediates (*Broglia et al., 2020*). To determine the extent to which these housekeeping nucleases catalyze crRNA maturation on their own, we performed nuclease assays with purified components (*Figure 2A*). In these assays, Cas10-Csm complexes loaded with 71 nt intermediate crRNAs (Cas10-Csm (71)) were purified from LM1680/Δ*rnr*Δ*pnp* (*Figure 2B*) and combined with purified RNase R, PNPase, and/or

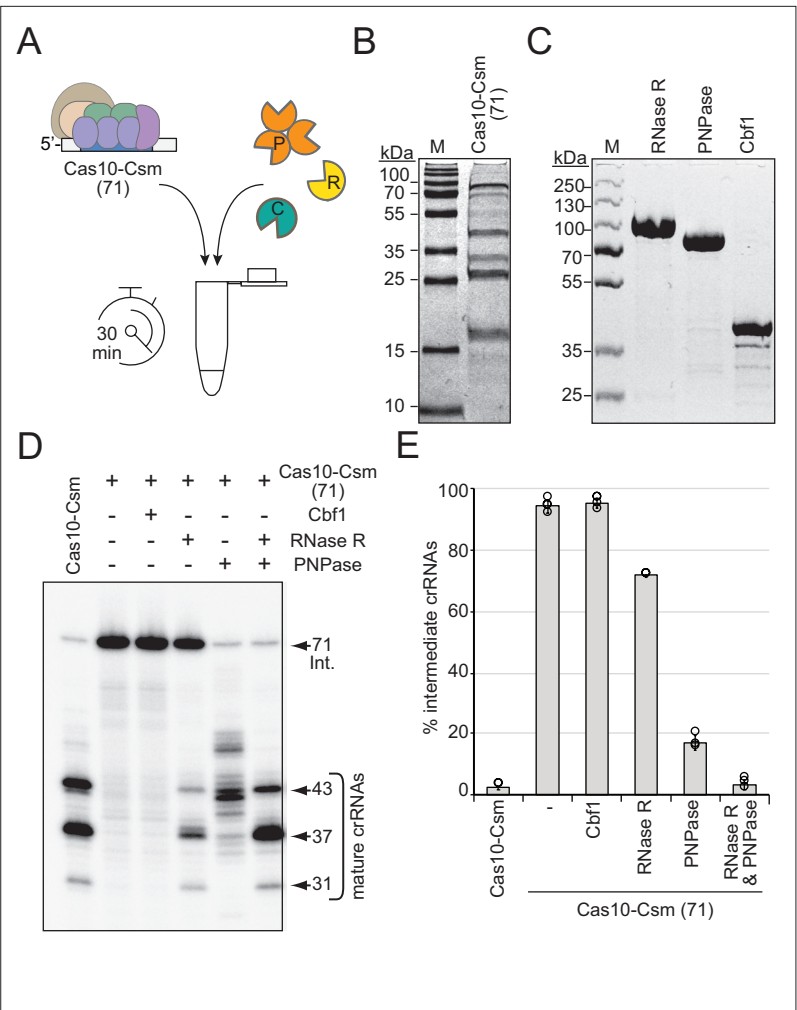

**Figure 2.** RNase R and PNPase are sufficient to complete crRNA maturation in a purified system. (**A**) Illustration of experimental flow of the crRNA maturation nuclease assay. P, PNPase; R, RNase R; C, Cbf1; Cas10-Csm (71), Cas10-Csm complexes purified from *S. epidermidis* LM1680Δ*pnp*Δ*rnr*. (**B**) Purified Cas10-Csm (71) complexes used in this assay. See *Figure 2—source data 1*. M, denaturing protein marker. kDa, kilodalton. (**C**) Purified recombinant exonucleases RNase R, PNPase, and Cbf1 used in this assay. See *Figure 2—source data 2*. (**D**) Cas10-Csm (71) complexes were incubated with indicated nucleases for 30 min at 37°C. After digestion, crRNAs were extracted from the complexes, radiolabeled at their 5′-ends, and resolved on a denaturing gel. The leftmost lane shows crRNAs extracted from Cas10-Csm complexes purified from WT cells as a control. See also *Figure 2—figure supplement 1* and *Figure 2—source data 3*. (**E**) Quantification of complex-bound intermediate crRNAs (relative to total crRNAs) following crRNA maturation assays. The data represent an average of 2–4 independent trials (± S.D). See *Figure 2—source data 4*.

The online version of this article includes the following source data and figure supplement(s) for figure 2:

**Source data 1.** Raw uncropped image for panel B.

**Source data 2.** Raw uncropped image for panel C.

**Source data 3.** Raw uncropped image for panel D.

**Source data 4.** Percent intermediate crRNAs for individual replicates in panel E.

**Figure supplement 1.** RNase R alone cannot complete crRNA maturation in a purified system.

**Figure supplement 1—source data 1.** Raw uncropped image.

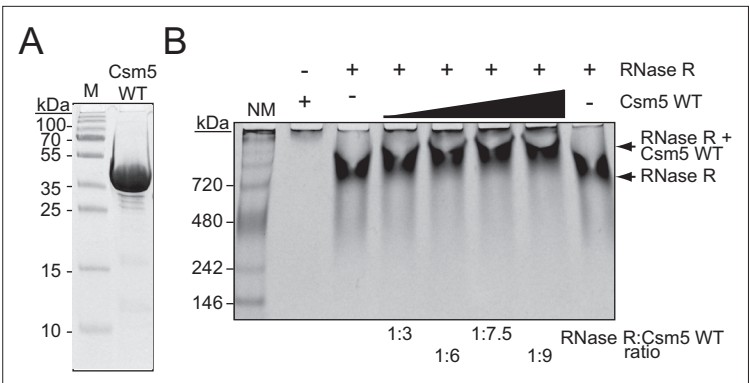

**Figure 3.** Csm5 interacts with RNase R. (**A**) Purified recombinant WT Csm5 is shown. The protein was resolved in an SDS-PAGE gel and visualized using Coomassie G-250 staining. M, denaturing protein marker; kDa, kilodalton. See *Figure 3—source data 1*. (**B**) Native gel showing RNase R resolved with increasing proportions of Csm5 WT. Shown is a representative of three independent trials. NM, native protein marker. See also *Figure 3—figure supplement 1*, *Figure 3—figure supplement 2*, and *Figure 3—source data 2*.

The online version of this article includes the following source data and figure supplement(s) for figure 3:

**Source data 1.** Raw uncropped image for panel A.

**Source data 2.** Raw uncropped image for panel B.

**Figure supplement 1.** Csm5 does not interact with bovine serum albumin (BSA).

**Figure supplement 1—source data 1.** Raw uncropped image.

**Figure supplement 2.** Csm5 interacts weakly with RNase R in a pulldown assay.

**Figure supplement 2—source data 1.** Raw uncropped image for panel B.

**Figure supplement 2—source data 2.** Raw uncropped image for panel C.

**Figure supplement 2—source data 3.** Raw uncropped image for panel D.

Cbf1 (*Figure 2C*). After 30 min of incubation with appropriate divalent metals, crRNAs were extracted from the complexes and visualized (*Figure 2D and E*). As expected, the Cas10-Csm (71) complex is unable to catalyze crRNA maturation on its own. Interestingly, Cbf1 is also incapable of cleaving intermediate crRNAs associated with the complex. In contrast, RNase R and PNPase each cause partial processing of crRNA intermediates on their 3'-ends, with cleavage patterns similar to those observed when one or the other nuclease is deleted from cells—addition of PNPase produces a pattern of crRNA lengths similar to that seen in LM1680/Δ*rnr* cells, and addition of RNase R into the reaction generates crRNA lengths similar to those extracted from LM1680/Δ*pnp* cells (compare *Figures 1E and 2D*). Even when given up to 60 min in the in vitro assay, RNase R alone is unable to process about half of the intermediate crRNAs (*Figure 2—figure supplement 1*). However, when RNase R and PNPase are combined in the reaction, the majority of crRNA intermediates are processed to the appropriate mature lengths (43, 37, and 31 nt) with proportions bearing a striking resemblance to those observed when crRNAs are purified from WT cells (*Figure 2D and E*). Taken together, our data demonstrate that RNase R and PNPase are both necessary and sufficient to process intermediate crRNAs associated with the Cas10-Csm complex to achieve their final mature lengths.

## Csm5 interacts with RNase R

We next considered the mechanism of RNase R recruitment to the Cas10-Csm complex. Previously, we showed that Csm5 (a member of the complex within the Cas7 group) directly interacts with PNPase in a purified system (*Walker et al., 2017*), and since deletion of *csm5* causes complete loss of crRNA maturation while allowing for the remainder of the complex to form (*Hatoum-Aslan et al., 2013*), we reasoned that RNase R recruitment is also likely to be facilitated by Csm5. To test this, RNase R and Csm5 were resolved alone and combined in a native polyacrylamide gel, which separates proteins on the basis of size and charge. As expected, Csm5 fails to enter into the gel at near-neutral pH due to its basic isoelectric point (pI, *Supplementary file 1*) and resulting positive charge in the native running conditions, while RNase R migrates into the native gel and shows up as a band following Coomassie

staining (*Figure 3A and B*). Consistent with previous observations, RNase R appears to self-associate in vitro (*Cheng and Deutscher, 2002*), which accounts for its shallow migration into the gel. Nonetheless, we observed that the addition of increasing amounts of Csm5 to RNase R causes the band to shift upward, indicating that the two proteins interact. This interaction is likely weak/transient because Csm5 must be added in excess (up to 9:1) to observe a noticeable band shift. To confirm that the interaction is specific to RNase R, we repeated the same assay using bovine serum albumin (BSA), which also has an acidic isoelectric point (*Supplementary file 1*), and no such shift was observed (*Figure 3—figure supplement 1*). To provide further support for a direct interaction between Csm5 and RNase R, we performed an affinity pulldown assay (*Figure 3—figure supplement 2A*). In this assay, Csm5-His10-Smt3 is loaded onto a Ni²⁺-agarose column, the column is washed to remove unbound protein, and then untagged RNase R (or protein buffer) is allowed to flow through the column. Following extensive washing of unbound proteins, proteins remaining in the column are eluted three times using a buffer containing imidazole. Consistent with the weak/transient interaction observed between the two proteins, non-stoichiometric amounts of RNase R were found to co-elute with Csm5 (*Figure 3—figure supplement 2B and C*). Importantly, untagged RNase R alone fails to stick to the column when subjected to the same wash and elution steps (*Figure 3—figure supplement 2D*). These data suggest that Csm5 facilitates recruitment of RNase R to the Cas10-Csm complex.

## Csm5 binds and stimulates PNPase through a predicted disordered region

Csm5 is about half the size of RNase R and PNPase (*Supplementary file 1*), and considering that PNPase functions as a trimer (*Symmons et al., 2000*), we wondered how Csm5 provides binding sites for both proteins. One possibility is that the nuclease docking site(s) might be spread over multiple subunits of the Cas10-Csm complex, with Csm5 contributing to the bulk of the interaction(s). Another nonexclusive possibility is that both nucleases may be recruited by the same/overlapping binding site(s) on Csm5, with one or the other allowed to occupy the site at any given time. Such transient and dynamic interactions are known to occur with proteins bearing intrinsically disordered regions (IDRs), flexible polypeptides enriched with charged residues that have the capacity to bind multiple partners (*Dyson and Wright, 2005*; *Chakrabarti and Chakravarty, 2022*; *Bigman et al., 2022*). Indeed, Csm5 is enriched with positively charged amino acids that confer its basic pI (*Figure 4A* and *Supplementary file 1*). Based on these observations, we hypothesized that Csm5 mediates binding of RNase R and/or PNPase via one or more IDR(s).

To begin to test this hypothesis, we searched for predicted disordered regions in *S. epidermidis* Csm5 using the web-based tool PONDR (Predictor of Natural Disordered Regions), which uses neural networks to discriminate between ordered and disordered residues in a given protein (*Romero et al., 2004*). This analysis revealed the presence of two putative disordered regions spanning residues 109–116 and 310–320 (here onward IDR1 and IDR2, respectively), with the latter having the higher probability for being disordered (*Figure 4—figure supplement 1A and B*). These regions were next examined in relation to the distribution of charged residues across Csm5, and we noted that both IDRs encompass multiple positively charged residues (*Figure 4A*). To gain a better understanding of the structural context of the predicted IDRs, we examined the homologous residues in the experimentally determined structure of the Cas10-Csm complex from *Streptococcus thermophilus* (StCas10-Csm) (*You et al., 2019*). Homologous residues in Csm5 were identified in a Clustal pairwise sequence alignment and mapped back to the StCsm5 structure. We found that the StCsm5 subunit in the unbound complex (PDB ID 6IFN) has half of its amino acids within loop/coil structures (*Figure 4—figure supplement 1C*, magenta), and the residues homologous to those comprising IDR2 in *S. epidermidis* Csm5 align well with a long loop structure in StCsm5 (*Figure 4—figure supplement 1C*, cyan). Further, while this article was under review, several cryo-EM structures of the Cas10-Csm complex from *S. epidermidis* were reported (*Smith et al., 2022*), and analysis of Csm5 in the unbound Cas10-Csm complex (PDB ID 7V02) revealed similar trends. Specifically, nearly half of the residues in Csm5 (~44%) reside in loop/coil structures or were unresolved (*Figure 4—figure supplement 1D*, magenta or not visible, respectively), and IDR 2 comprises a short loop and a short beta strand (*Figure 4—figure supplement 1D* D, cyan). Notably, IDR2 lies directly adjacent to 19 residues that are unresolved in the structure (amino acids 291–309). These observations lend support to the notion that Csm5 may possess one or more IDRs, which play role(s) in nuclease recruitment.

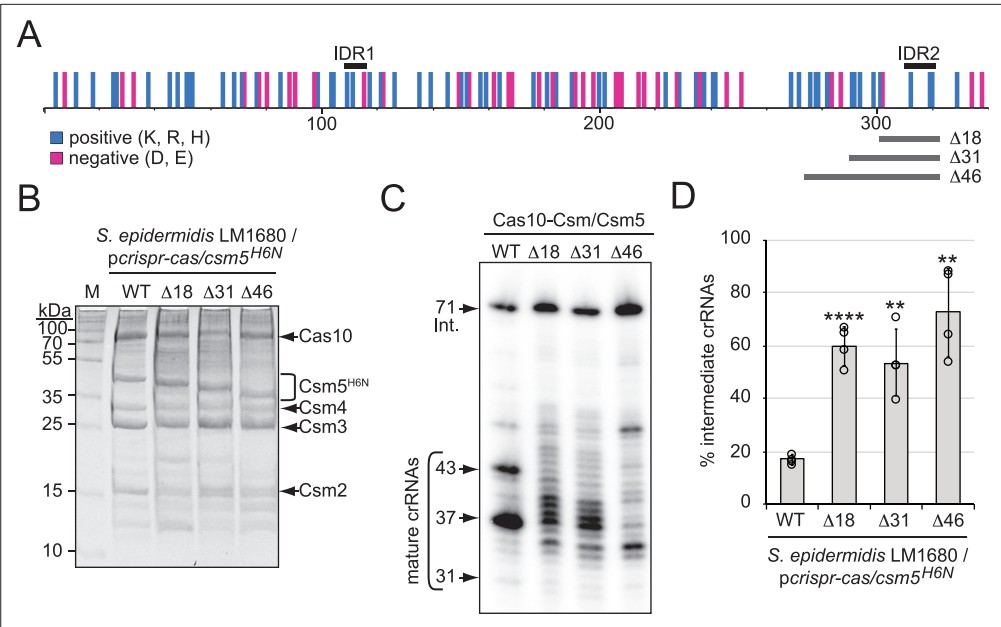

**Figure 4.** A predicted disordered region in Csm5 promotes crRNA maturation. (**A**) Illustration showing the distribution of charged residues, predicted disordered regions, and truncations introduced in Csm5. Positions of charged residues (positive, cyan; negative, magenta) are shown as vertical bars. Predicted intrinsically disordered regions (IDR1 and IDR2) and regions that were truncated are delimited by black and gray horizontal bars above and below, respectively. K, lysine; R, arginine; H, histidine; D, aspartate; E, glutamate. See also *Figure 4—figure supplement 1*. (**B**) Cas10-Csm complexes with various Csm5 truncations are shown. Complexes were extracted from *S. epidermidis* LM1680 cells harboring p*crispr-cas/csm5*[H6N], which has a 6-His tag on the N-terminus of Csm5 to confirm full complex assembly. Complexes were purified using Ni²⁺ affinity chromatography, resolved on and SDS-PAGE gel, and visualized with Coomassie G-250 staining. M, denaturing protein marker; kDa, kilodalton. See also *Figure 4—source data 1*. (**C**) Total crRNAs bound to indicated Cas10-Csm complexes were extracted, radiolabeled at their 5′-ends, and resolved on a denaturing gel. See also *Figure 4—source data 2*. (**D**) Fractions of complex-bound intermediate crRNAs relative to total crRNAs are shown for Csm5 truncation mutants. The percent of intermediate crRNAs represents the ratio of the intermediate (71 nt) band density to the sum of band densities of the major crRNA species (71, 43, 37, and 31 nt). The data represents an average of four independent trials (± S.D). A two-tailed *t*-test was performed to determine significance and p-values obtained were <0.005 (**) or <0.00005 (****). See also *Figure 4—source data 3*.

The online version of this article includes the following source data and figure supplement(s) for figure 4:

**Source data 1.** Raw uncropped image for panel B.

**Source data 2.** Raw uncropped image for panel C.

**Source data 3.** Percent intermediate crRNAs for individual replicates in panel D.

**Figure supplement 1.** Predicted disordered regions of Csm5.

---

To further test this hypothesis, we deleted the regions encoding IDR1 and IDR2 from *csm5* in a plasmid bearing the entire CRISPR-Cas system of *S. epidermidis* RP62a (p*crispr-cas/csm5*[H6N]) (*Hatoum-Aslan et al., 2013*). Importantly, the 6-His tag in this construct is located on Csm5 to allow us to rule out mutations that impact Csm5 stability and/or complex assembly. The plasmids were introduced into *S. epidermidis* LM1680 and cell lysates were subjected to Ni²⁺-affinity chromatography. We were unable to purify complexes when the IDR1 region in Csm5 was deleted (not shown), indicating that the residues comprising IDR1 might be important for Csm5 stability and/or complex assembly. In contrast, full complexes were recovered in the presence of three deletions spanning 18, 31, or 46 amino acids encompassing IDR2 (*Figure 4A and B*). Interestingly, crRNAs bound to these complexes exhibited a range of aberrant lengths with significant (>50%) accumulation of 71 nt intermediates (*Figure 4C and D*), suggesting that IDR2 within Csm5 may facilitate interactions with RNase R and/or PNPase.

We further tested for direct interactions using gel shift assays, in which Csm5Δ46 was purified (*Figure 5A*), combined with RNase R or PNPase in different proportions, and resolved on native

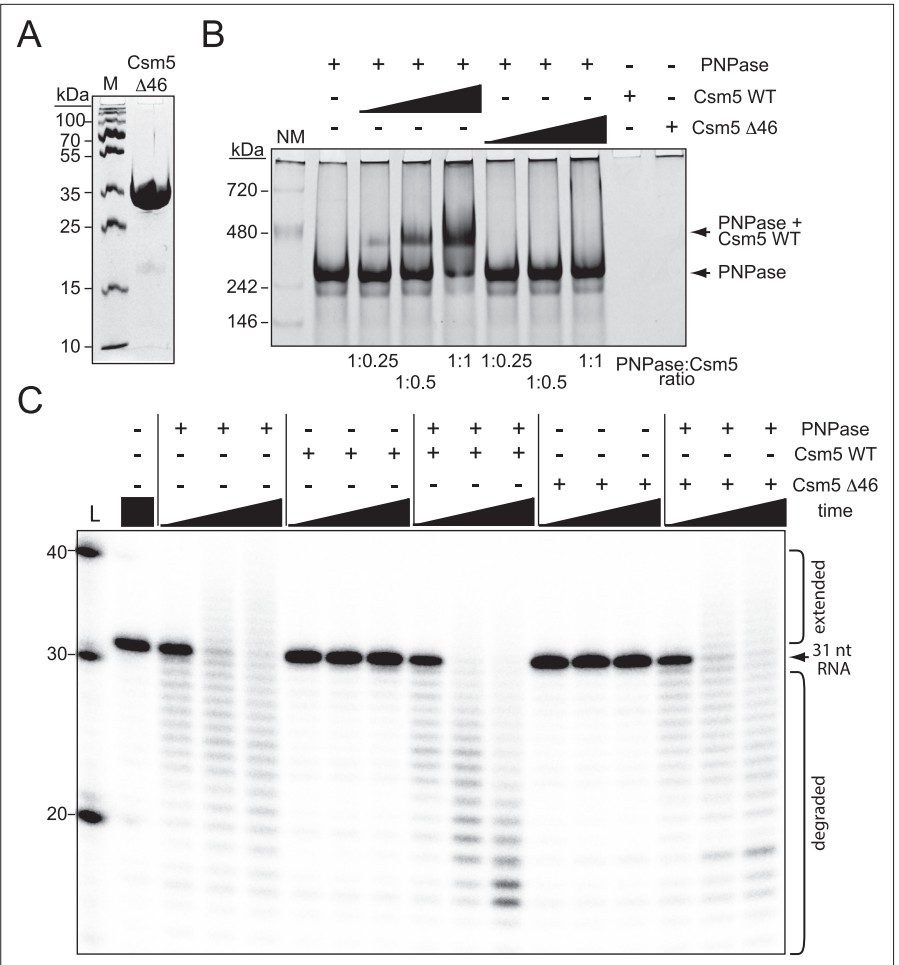

**Figure 5.** Csm5 interacts with and stimulates PNPase via a predicted disordered region. (**A**) Purified recombinant Csm5Δ46 is shown, in which IDR2 has been deleted. The protein was resolved on an SDS-PAGE gel and visualized using Coomassie G-250 staining. M, denaturing protein marker; kDa, kilodalton. See also *Figure 5—source data 1*. (**B**) PNPase was resolved on a native gel with increasing amounts of Csm5 (WT and Δ46). Shown is a representative of three independent trials. NM, native protein marker. See also *Figure 5—source data 2*. (**C**) Nuclease assays conducted with PNPase and/or Csm5 (WT and Δ46) are shown. In these assays, a 5'-end labeled 31-nucleotide RNA substrate was combined with indicated proteins, incubated at 37°C for increasing amounts of time (0.5, 5, and 15 mins), and resolved on a denaturing gel. Shown is a representative of two independent trials. L, RNA Ladder. See also *Figure 5—source data 3*.

The online version of this article includes the following source data and figure supplement(s) for figure 5:

**Source data 1.** Raw uncropped image for panel A.

**Source data 2.** Raw uncropped image for panel B.

**Source data 3.** Raw uncropped image for panel C.

**Figure supplement 1.** Csm5Δ46 retains interaction with RNase R.

**Figure supplement 1—source data 1.** Raw uncropped image.

polyacrylamide gels. The results showed that while Csm5Δ46 maintains its interaction with RNase R (*Figure 5—figure supplement 1*), the mutant has little/no interaction with PNPase (*Figure 5B*), indicating that the IDR2 region is essential for PNPase binding in vitro. Previously, we showed that in addition to the physical interaction between Csm5 and PNPase, there is also a functional interaction in which Csm5 stimulates PNPase's nucleolytic activity (*Walker et al., 2017*). To determine the impact of IDR2 on PNPase activity, nuclease assays were performed in which a 31 nt ssRNA substrate was incubated with PNPase and/or Csm5 for increasing amounts of time. Consistent with previous results, we found that Csm5 alone has no impact on substrate length, PNPase (which is a dual polymerase

nuclease) causes both RNA extension and degradation, and when the two proteins are combined, stimulation of PNPase's nucleolytic activity occurs (*Figure 5C*). Importantly, Csm5Δ46 fails to cause such stimulation, consistent with the loss of physical interaction between Csm5Δ46 and PNPase. Taken together, these results suggest that the IDR2 region of Csm5 likely plays a role in the recruitment and stimulation of PNPase, while the binding site for RNase R may reside elsewhere in Csm5.

## RNase R and PNPase work synergistically to maintain robust anti-plasmid immunity

We next wondered about the extent to which RNase R and/or PNPase impact type III CRISPR-Cas function. We began by testing immunity against diverse staphylococcal phages using various over-expression systems (*Figure 6—figure supplement 1*). First, immunity against siphovirus CNPx was tested in *S. epidermidis* LM1680 bearing p*crispr-cas/csm2*^H6N (*Figure 6—figure supplement 1A*). In this plasmid, the second spacer (*spc2*) targets the phage pre-neck appendage (*cn20*) gene (*Daniel et al., 2007*), which is likely to be expressed late in the phage infection cycle. Phage challenge assays were performed by spotting tenfold dilutions of CNPx atop lawns of LM1680 cells bearing variants of the plasmid, incubating plates overnight, and enumerating phage plaques (i.e., clear zones of bacterial death) the next day. As expected, lawns of WT cells with the WT plasmid showed zero plaques, while lawns of WT cells bearing the empty vector allowed for tens of millions of plaques to form (measured as plaque-forming units per milliliter [pfu/ml]) (*Figure 6—figure supplement 1B*). Interestingly, deletion of *rnr* alone or in combination with *pnp* caused no detectable defect in immunity. Surprisingly, even deletion of *csm5* from the plasmid had no impact on CRISPR function. In light of these observations, we wondered whether overexpressing the CRISPR-Cas system might compensate for mild defects. Thus, we tested another system that relies upon *S. epidermidis* RP62a (with an intact *crispr-cas* locus) bearing the multicopy plasmid p*crispr*, which contains a single repeat and spacer targeting phage(s) of interest (*Bari et al., 2017*). Since CNPx cannot form plaques on RP62a, phage challenge assays with podophage Andhra and myophage ISP were performed (*Figure 6—figure supplement 1C–F*). Consistent with previous observations, the WT strain with the empty vector allows for the formation of millions-billions of plaques, while RP62a strains with p*crispr* and appropriate phage-targeting spacers allow for zero plaques to form. Interestingly, RP62a strains devoid of *rnr* and/or *pnp* maintained robust immunity against both phages. Since previous work has shown that type III-A immunity relies more heavily on the accessory nuclease Csm6 when phage late gene(s) are targeted owing to the lag in transcript/protospacer expression (*Jiang et al., 2016*), we explored the impact of targeting genes that are predicted to be expressed early vs. late in the infection cycle. Specifically, spacers targeting genes that encode Andhra's DNA polymerase (early, *spcA1*), major tail protein (late, *spcA2*), and lysin-like peptidase (late, *spcA3*), as well as ISP's lysin (late, *spcI*), were tested. Contrary to our expectations, robust anti-phage immunity was maintained in all strains. These results indicate that RNase R and PNPase may have little/no impact on immunity against diverse phages in these overexpression systems.

We next tested anti-plasmid immunity using a conjugation assay that relies entirely upon chromosomally encoded components (*Figure 6A and B*). In this assay, *S. epidermidis* RP62a recipients are mated with *S. aureus* RN4220 cells harboring the conjugative plasmid pG0400. The first spacer in RP62a's *crispr-cas* locus (*spc1*) bears complementarity to the nickase (*nes*) gene in pG0400 and therefore mitigates the conjugative transfer of the plasmid (*Marraffini and Sontheimer, 2008*). Consistent with previous observations, mating assays performed with *S. aureus* RN4220/pG0400-WT donor cells and *S. epidermidis* RP62a WT recipients produced hundreds of transconjugants (i.e., *S. epidermidis* recipients that have acquired pG0400-WT); however, when *S. epidermidis* RP62a/Δ*spc1-3* cells were used as recipients, >10,000 transconjugants were recovered (*Figure 6C*). Interestingly, while RP62a/Δ*pnp* performed similarly to the WT strain, RP62a/Δ*rnr* exhibited a moderate attenuation in immunity, as evidenced by a significantly higher conjugation efficiency compared to that of WT (*Figure 6—source data 1*). This defect was absent in the complemented strain (RP62a/Δ*rnr::rnr**), confirming that deletion of *rnr* is indeed responsible for the phenotype. Strikingly, the double mutant (RP62a/Δ*rnr*Δ*pnp*) showed a near complete loss of immunity, while the complemented strain RP62a/Δ*rnr*Δ*pnp::rnr** performed similarly to WT and RP62a/Δ*pnp*. These data demonstrate that RNase R and PNPase work synergistically to promote anti-plasmid immunity.

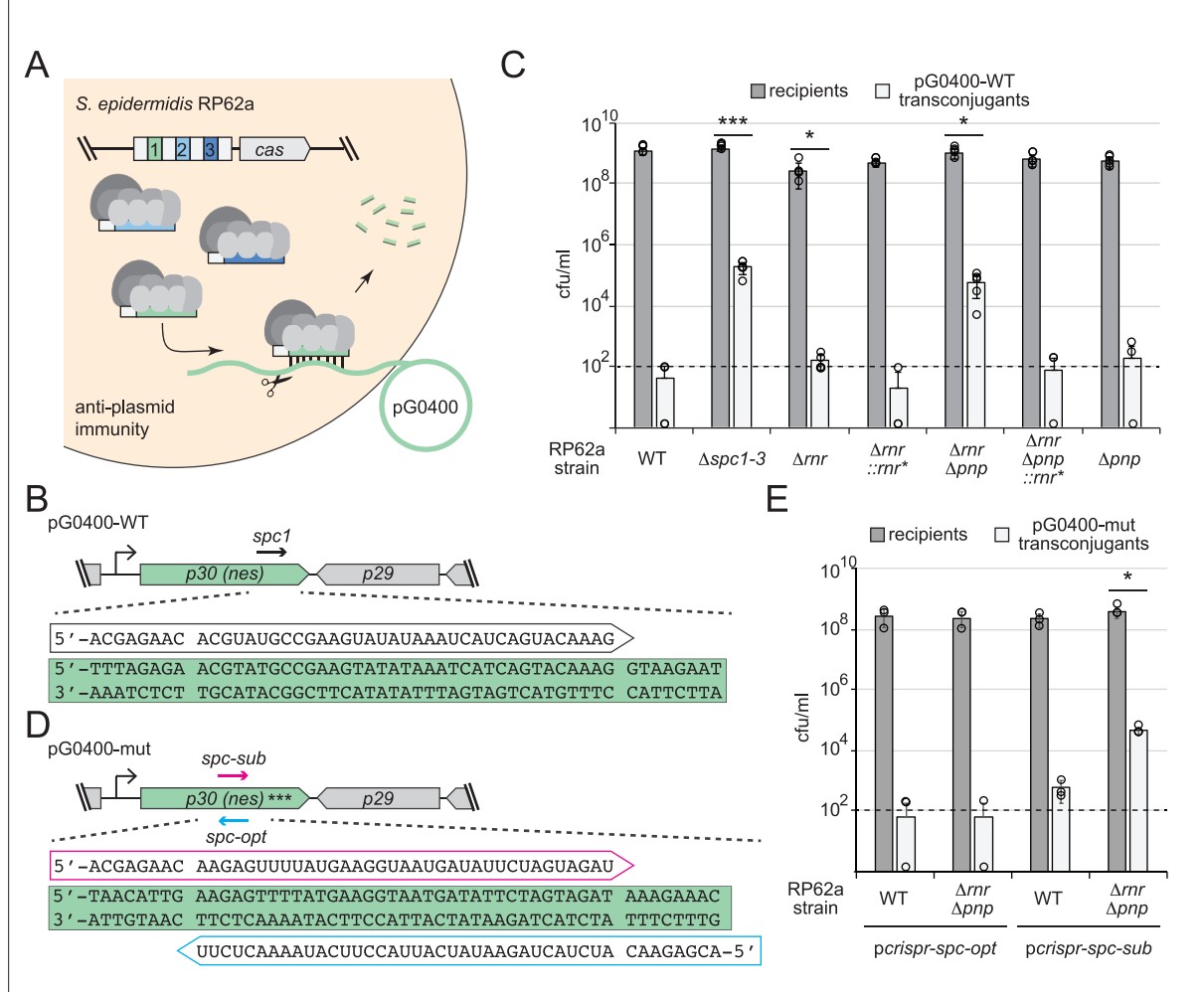

**Figure 6.** RNase R and PNPase work synergistically to promote robust anti-plasmid immunity. (**A**) Illustration of the anti-plasmid assay is shown in which the conjugative plasmid pG0400 is transferred from a *S. aureus* RN4220 donor (not shown) into various *S. epidermidis* RP62a recipient strains. The first spacer in the CRISPR locus (green square) bears complementarity to the nickase (*nes*) gene in pG0400. (**B, D**) Sequences of protospacers and corresponding crRNAs targeting pG0400-WT (**B**) and pG0400-mut (**D**). Protospacer sequences are highlighted in green, and targeting crRNA sequences are shown in unfilled arrows. In pG0400-mut, asterisks represent nine silent mutations in the *spc1* protospacer region. (**C**) Results from conjugation assays in which indicated *S. epidermidis* RP62a recipient strains were mated with *S. aureus* RN4220/pG0400-WT donor cells. See *Figure 6—source data 1*. (**E**) Results from conjugation assays in which various *S. epidermidis* RP62a recipient strains harboring indicated plasmids were mated with *S. aureus* RN4220/pG0400-mut donor cells. See *Figure 6—source data 2*. In panels (**C**) and (**E**), numbers of recipients and transconjugants following mating are shown in cfu/ml (colony-forming units per milliliter). Graphs show an average of five (**C**) or three (**E**) independent trials (± SD). Individual data points are shown with open circles, and data points on the x-axis represent at least one replicate where a value of 0 was obtained. The dotted line indicates the limit of detection for this assay. Two-tailed *t*-tests were performed on conjugation efficiencies to determine significance, and p-values of <0.05 (*) or <0.0005 (***) were obtained.

The online version of this article includes the following source data and figure supplement(s) for figure 6:

**Source data 1.** Recipients, transconjugants, and conjugation efficiencies for independent replicates in panel C.

**Source data 2.** Recipients, transconjugants, and conjugation efficiencies for independent replicates in panel E.

**Figure supplement 1.** RNase R and PNPase are dispensable for anti-phage immunity.

**Figure supplement 1—source data 1.** Phage plaque counts for individual replicates in panel B.

**Figure supplement 1—source data 2.** Phage plaque counts for individual replicates in panel D.

**Figure supplement 1—source data 3.** Phage plaque counts for individual replicates in panel F.

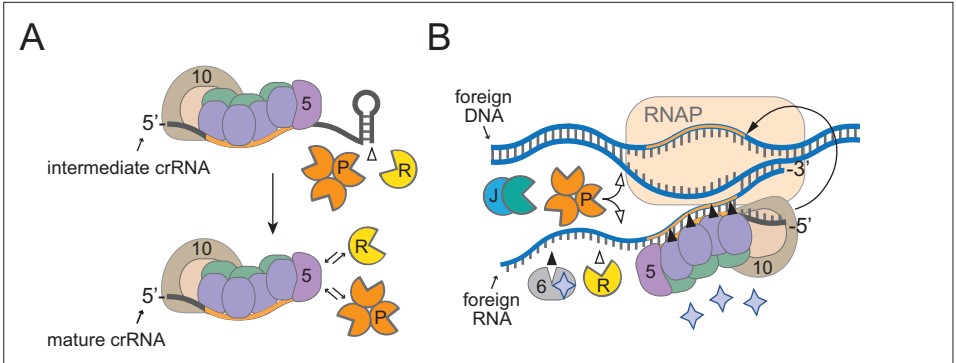

**Figure 7.** A model for how diverse housekeeping nucleases are enlisted to ensure successful defense. (**A**) During Cas10-Csm complex assembly, Csm5 recruits and/or stimulates RNase R and PNPase through direct interactions. The unprotected 3'-ends of intermediate crRNAs are trimmed as a consequence of nuclease recruitment, resulting in the generation of the shorter mature species. (**B**) During interference, RNase R and PNPase work synergistically to help degrade invading nucleic acids alongside other Cas and non-Cas nucleases. Filled triangles illustrate events catalyzed by Cas proteins, and open triangles illustrate events catalyzed by non-Cas nucleases. Purple stars represent cyclic oligoadenylate molecules produced by Cas10. 5, Csm5; 6, Csm6; 10, Cas10; R, RNase R; P, PNPase; J, RNase J1/J2; RNAP, RNA polymerase.

Although *spc1* was naturally acquired in *S. epidermidis* RP62a and promotes anti-plasmid immunity in the WT background, it can be considered suboptimal because the crRNA that it encodes has the same (not complementary) sequence as the *nes* transcript (*Figure 6B*). Accordingly, *spc1*-mediated anti-plasmid immunity was found to rely upon recognition of sparse antisense transcripts presumably originating from a weak promoter downstream of *nes* (*Rostøl and Marraffini, 2019*). In light of these observations, we wondered whether the defect in anti-plasmid immunity in RP62a/*ΔrnrΔpnp* could be alleviated by targeting the more abundant *nes* transcript. To test this, we designed two spacers against the *nes* open-reading frame, *spc-opt* and *spc-sub*, which encode crRNAs that are complementary to (optimal) and identical to (suboptimal) the *nes* transcript, respectively (*Figure 6D*). Importantly, these spacers completely overlap and therefore share the same GC content. Furthermore, the corresponding protospacers are devoid of complementarity between the 5'-tag on the crRNA and corresponding anti-tag region on the targeted transcript, a necessary condition that signals 'non-self' and licenses immunity (*Marraffini and Sontheimer, 2010*). These spacers were inserted into p*crispr* to create p*crispr-spc-opt* and p*crispr-spc-sub*, and the plasmids were introduced into RP62a WT and RP62a/*ΔrnrΔpnp*. In order to eliminate the effects of the chromosomally encoded *spc1* in these strains, conjugation assays were performed with *S. aureus* RN4220 cells bearing pG0400-mut, a variant of pG0400-WT that is not recognized by the *spc1* crRNA owing to the presence of nine silent mutations across the protospacer (*Marraffini and Sontheimer, 2008*). We found that similarly to *spc1*, *spc-sub* mediates anti-plasmid immunity in RP62a WT, but fails to do so in RP62a/*ΔrnrΔpnp* (*Figure 6E*). In contrast, *spc-opt* facilitates robust immunity in both WT and the double mutant. Altogether, these data support the notion that the nuclease activities of RNase R and PNPase are essential to maintain robust immunity when targeted transcripts are in low abundance.

## Discussion

Here, we elucidate the complete pathway for crRNA maturation in a model type III-A CRISPR-Cas system and expand the repertoire of known accessory nucleases required for immunity (*Figure 7*). Most functionally characterized type III systems generate mature crRNAs that vary in length on their 3'-ends by 6 nt increments (*Hale et al., 2009*; *Hatoum-Aslan et al., 2011*; *Staals et al., 2013*; *Tamulaitis et al., 2014*), and, while this periodic cleavage pattern is known to derive from the protection offered by variable copies of Csm3/Cmr4 subunits within effector complexes (*Hatoum-Aslan et al., 2013*; *You et al., 2019*; *Osawa et al., 2015*; *Dorsey et al., 2019*), the identity of the nuclease(s) responsible for crRNA 3'-end maturation and the functional significance of this additional processing step have long remained a mystery. Here, we demonstrate that two housekeeping nucleases, RNase R and PNPase, work in concert to trim the 3'-ends of intermediate crRNAs (*Figures 1 and 2*) and

promote robust anti-plasmid immunity in *S. epidermidis* (*Figure 6*). Since intermediate crRNAs can mediate a successful immune response under certain conditions, such as when the CRISPR-Cas system is overexpressed (*Figure 6—figure supplement 1*) or when a highly expressed transcript is targeted (*Figure 6D and E*), the functional value of crRNA maturation is likely nominal and may occur simply as a consequence of the preemptive recruitment of RNase R and PNPase to the effector complex to assist during interference.

It is now well understood that most, if not all, type III CRISPR-Cas systems rely upon RNA recognition to eliminate invading DNA (*Samai et al., 2015*; *Kazlauskiene et al., 2016*; *Estrella et al., 2016*; *Elmore et al., 2016*; *Liu et al., 2017*), and this feature presents unique challenges owing to the fact that targeted transcripts can have variable levels of expression. Previous studies have shown that while highly expressed target RNAs elicit a robust and sustained immune response that results in the swift elimination of nucleic acid invaders, low-abundance targets evoke a weak immune response, and corresponding invaders are more difficult to clear (*Goldberg et al., 2014*; *Jiang et al., 2016*; *Rostøl and Marraffini, 2019*). In the latter scenario, the Cas10-Csm complex requires the help of Csm6, an accessory nuclease encoded in the CRISPR-Cas locus that is not part of the complex. Csm6 has nonspecific endoribonuclease activity that is stimulated when bound to cOAs produced by Cas10 (*Kazlauskiene et al., 2017*; *Niewoehner et al., 2017*; *Nasef et al., 2019*), and Csm6-mediated degradation of transcripts derived from both the invader and host causes growth arrest until the foreign nucleic acids are cleared (*Rostøl and Marraffini, 2019*). Once recruited by the complex, PNPase and RNase R likely degrade nucleic acids in the vicinity nonspecifically, similarly to Csm6. Interestingly, Csm6 is dispensable for immunity when targeted transcripts are highly expressed (*Jiang et al., 2016*; *Rostøl and Marraffini, 2019*), similar to RNase R and PNPase (*Figure 6*). Altogether, our observations support a model in which RNase R and PNPase are recruited as accessory nucleases to ensure a successful defense against nucleic acid invaders, particularly when targeted transcripts have low abundance (*Figure 7*).

In spite of these similarities with Csm6, RNase R and PNPase have distinct functional roles in the cell and mechanisms by which they are enlisted for defense. Unlike Csm6, PNPase and RNase R are 3′–5′ exonucleases primarily involved in housekeeping functions—PNPase is a member of the RNA degradosome, a multi-enzyme complex that catalyzes RNA processing and degradation, and RNase R performs similar/overlapping functions, but works independently of the degradosome in most organisms (*Bechhofer and Deutscher, 2019*; *Tejada-Arranz et al., 2020*). In addition to its RNase activity, PNPase has the capacity to degrade single-stranded DNA (*Walker et al., 2017*), presumably to facilitate DNA repair (*Cardenas et al., 2009*). These activities are harnessed by the Cas10-Csm complex through direct interactions—Csm5 essentially borrows both nucleases through weak/transient interactions, and PNPase's nuclease activity is further stimulated when bound to Csm5 (*Figures 3 and 5*). While the specific binding site for RNase R remains unknown, a predicted IDR on the C-terminus of Csm5 is responsible for recruitment and stimulation of PNPase (*Figures 4 and 5*, *Figure 4—figure supplement 1*, *Figure 5—figure supplement 1*). Since RNase R and PNPase are each about twice the size of Csm5, their association with the Cas10-Csm complex is likely mutually exclusive, and their docking site(s) may also overlap with other components in the complex. Regardless of the precise molecular requirements for recruitment, Cas10-Csm's ability to interface with diverse cellular nucleases is remarkable and bears striking parallels to the assembly of prokaryotic degradosomes and eukaryotic granules, which rely upon 'hub' proteins to recruit multiple members of enzyme complexes through transient interactions with one or more IDRs (*Tejada-Arranz et al., 2020*). Given that most type III CRISPR-Cas systems possess Csm5 homologs (*Makarova et al., 2020b*), and RNase R and PNPase are evolutionarily conserved in prokaryotes and eukaryotes (*Zuo and Deutscher, 2001*), their enlistment in type III CRISPR-Cas defense may be a common feature in diverse organisms.

## Ideas and speculation

Harnessing the activities of housekeeping nucleases and channeling their diverse activities toward defense may have evolved as a strategy to minimize the genetic footprint of complex immune systems while cutting the energetic costs associated with manufacturing enzymes with redundant functions. Supporting this notion, diverse CRISPR-Cas systems have been shown to tap into the pool of cellular housekeeping nucleases to perform different steps in their immunity pathways. The earliest example of this phenomenon was discovered over a decade ago in the type II CRISPR-Cas system

of *Streptococcus pyogenes*, in which processing of both crRNAs and the trans-encoded small RNA (tracrRNA) was shown to rely upon RNase III (*Deltcheva et al., 2011*). RNase III-mediated crRNA/tracrRNA processing is now considered a universal feature of type II systems (*Makarova et al., 2020b*). In addition, new spacer acquisition (i.e., adaptation) in type I and II systems has been shown to rely upon the DNA repair machinery RecBCD and AddAB in Gram-negative and -positive organisms, respectively (*Levy et al., 2015*; *Modell et al., 2017*). Also, we showed that RNase J2 plays a critical role in interference in the type III-A system of *S. epidermidis* (*Chou-Zheng and Hatoum-Aslan, 2019*). Beyond these more common systems, CRISPR-Cas variants in which one or more *cas* nucleases are missing were shown to rely upon degradosome nucleases to perform essential functionalities. In one such example, a type III-B variant in *Synechocystis* 6803 that lacks a Cas6 homolog relies upon RNase E to catalyze processing of pre-crRNAs (*Behler et al., 2018*). In another more extreme example, a unique CRISPR element in *Listeria monocytogenes*, which is completely devoid of *cas* genes, was shown to utilize PNPase for crRNA processing and interference (*Sesto et al., 2014*). Since housekeeping nucleases are needed on a daily basis and therefore less likely to be lost via natural selection, it is plausible that their enlistment in nucleic acid defense may be more widespread than currently appreciated.

# Materials and methods

## Key resources table

| Reagent type (species) or resource | Designation | Source or reference | Identifiers | Additional information |
|---|---|---|---|---|
| Gene (*Staphylococcus epidermidis*) | cbf1 | NA | GenBank: CP000029.1_SERP1378 | Encodes Cbf1 |
| Gene (*S. epidermidis*) | rnr | NA | GenBank: CP000029.1_SERP0450 | Encodes RNase R |
| Gene (*S. epidermidis*) | pnp | NA | GenBank: CP000029.1_SERP0841 | Encodes PNPase |
| Gene (*S. epidermidis*) | csm5 | NA | GenBank: CP000029.1_SERP2457 | Encodes Csm5 |
| Strain, strain background (*Staphylococcus aureus*, RN4220) | RN4220 | PMID:21378186 | GenBank: NZ_AFGU00000000 | LA Marraffini (Rockefeller University) |
| Strain, strain background (*S. epidermidis*, RP62a) | RP62a | PMID:3679536 | GenBank: CP000029.1 | LA Marraffini (Rockefeller University) |
| Strain, strain background (*S. epidermidis*, LAM104) Δspc1-3 | PMID:19095942 | | | LA Marraffini (Rockefeller University), derivative of RP62a with *crispr* deletion |
| Strain, strain background (*S. epidermidis*, LM1680) LM1680 | PMID:24086164 | | | LA Marraffini (Rockefeller University), derivative of RP62a with large deletion |
| Strain, strain background (phage Andhra) | Andhra | PMID:28357414 | GenBank: KY442063 | Isolated in-house |
| Strain, strain background (phage ISP) | ISP | PMID:21931710 | GenBank: FR852584 | LA Marraffini (Rockefeller University) |
| Strain, strain background (phage CNPx) | CNPx | PMID:26755632 | GenBank: NC_031241 | LA Marraffini (Rockefeller University) |
| Genetic reagent (*Staphylococcus epidermidis*, RP62a) | RP62a Δpnp | PMID:30942690 | | Created in-house |
| Genetic reagent (*S. epidermidis*, RP62a) | RP62a Δrnr | This paper | | The central 2316 nucleotides of the *rnr* coding region are deleted, see *Figure 1—figure supplement 1* |
| Genetic reagent (*S. epidermidis*, RP62a) | RP62a Δrnr::rnr* | This paper | | A copy of *rnr* with two silent mutations reintroduced into the *rnr* locus, see *Figure 1—figure supplement 1* |
| Genetic reagent (*S. epidermidis*, RP62a) | RP62a Δrnr Δpnp | This paper | | Contains in-frame deletions of *rnr* (described in the cells above) and *pnp* (PMID:30942690) |
| Genetic reagent (*S. epidermidis*, RP62a) | RP62a Δrnr Δpnp::rnr* | This paper | | A copy of *rnr* with two silent mutations reintroduced into the *rnr* locus, see *Figure 1—figure supplement 1* |
| Genetic reagent (*S. epidermidis*, LM1680) | LM1680 Δpnp | PMID:30942690 | | Created in-house |

*Continued on next page*

*Continued*

| Reagent type (species) or resource | Designation | Source or reference | Identifiers | Additional information |
|---|---|---|---|---|
| Genetic reagent (*S. epidermidis*, LM1680) | LM1680 Δ*rnr* | This paper | | The central 2316 nucleotides of the *rnr* coding region are deleted, see *Figure 1—figure supplement 1* |
| Genetic reagent (*S. epidermidis*, LM1680) | LM1680 Δ*rnr::rnr\** | This paper | | A copy of *rnr* with two silent mutations reintroduced into the *rnr* locus, see *Figure 1—figure supplement 1* |
| Genetic reagent (*S. epidermidis*, LM1680) | LM1680 Δ*rnr* Δ*pnp* | This paper | | Contains in-frame deletions of *rnr* (described in the cells above) and *pnp* (PMID:30942690) |
| Genetic reagent (*S. epidermidis*, LM1680) | LM1680 Δ*rnr* Δ*pnp::rnr\** | This paper | | A copy of *rnr* with two silent mutations reintroduced into the *rnr* locus, see *Figure 1—figure supplement 1* |
| Recombinant DNA reagent | pKOR1 | PMID:16051359 | | LA Marraffini (Rockefeller University) |
| Recombinant DNA reagent | pKOR1-Δ*rnr* | This paper | | To create in-frame deletion of *rnr* via allelic replacement |
| Recombinant DNA reagent | pKOR1-*rnr\** | This paper | | To create complementation of *rnr* with silent mutations via allelic replacement |
| Recombinant DNA reagent | p*crispr-cas* | PMID:23935102 | | LA Marraffini (Rockefeller University) |
| Recombinant DNA reagent | pcrisprcas/csm5$^{H6N}$Δ18 | This paper | | Contains 18 amino acids deletion encompassing IDR2 in Csm5 |
| Recombinant DNA reagent | pcrisprcas/csm5$^{H6N}$Δ31 | This paper | | Contains 31 amino acids deletion encompassing IDR2 in Csm5 |
| Recombinant DNA reagent | pcrisprcas/csm5$^{H6N}$Δ46 | This paper | | Contains 46 amino acids deletion encompassing IDR2 in Csm5 |
| Recombinant DNA reagent | pET28b-H$_{10}$Smt3-*csm5* | PMID:28204542 | | Created in-house |
| Recombinant DNA reagent | pET28b-H$_{10}$Smt3-*csm5Δ46* | This paper | | Contains 46 amino acids deletion encompassing IDR2 in Csm5; for overexpression and purification of Csm5Δ46 |
| Sequence-based reagent | 5'-ACGAGAACAC GUAUGCCGA AGUAUAUAAAUC | Eurofins MWG Operon | | A 31-nt single-stranded RNA substrate for nuclease assays, see *Figure 5* |
| Sequence-based reagent | DNA oligonucleotides (multiple) | Eurofins MWG Operon | | To build and sequence recombinant DNA constructs, see *Supplementary file 2* |
| Sequence-based reagent | Decade Markers System | Fisher Scientific | Cat# AM7778 | |
| Peptide, recombinant protein | EcoRI | New England Biolabs | Cat# R0101S | |
| Peptide, recombinant protein | T4 Polynucleotide kinase | New England Biolabs | Cat# M0201L | |
| Peptide, recombinant protein | T4 DNA Ligase | New England Biolabs | Cat# M0202S | |
| Peptide, recombinant protein | DpnI | New England Biolabs | Cat# R0176S | |
| Peptide, recombinant protein | Lysostaphin | AmbiProducts via Fisher | Cat# NC0318863 | |
| Peptide, recombinant protein | Pierce Protease and Phosphatase Inhibitor Mini Tablets | Thermo Fisher | Cat# 88669 | |
| Peptide, recombinant protein | SUMO Protease | MCLAB, http://www.mclab.com/SUMO-Protease.html | Cat# SP-100 | |
| Peptide, recombinant protein | Bovine serum albumin (BSA) | VWR | Cat# 97061-420 | |
| Peptide, recombinant protein | PageRuler Plus Prestained Protein Ladder, 10–250 kDa | Thermo Fisher | Cat# 26619 | |

*Continued on next page*

*Continued*

| Reagent type (species) or resource | Designation | Source or reference | Identifiers | Additional information |
|---|---|---|---|---|
| Peptide, recombinant protein | NativeMark Unstained Protein Standard | Invitrogen via Thermo Fisher | Cat# LC0725 | |
| Commercial assay or kit | EZNA Cycle Pure Kit | Omega Bio-tek via VWR | Cat# 101318-892 | |
| Commercial assay or kit | EZNA Plasmid DNA Mini Kit | Omega Bio-tek via VWR | Cat# 101318-898 | |
| Commercial assay or kit | Advance Centrifugal Devices 10K MWCO | Pall via VWR | Cat# 89131-980 | |
| Commercial assay or kit | Disposable Gravity Flow Columns for Protein Purification | G-Biosciences via VWR | Cat# 82021-346 | |
| Commercial assay or kit | G-25 Spin Columns | IBI Scientific via VWR | Cat# IB06010 | |
| Chemical compound or drug | HisPur Ni-NTA Resin | Thermo Fisher | Cat# 88222 | |
| Chemical compound or drug | TRIzol Reagent | Thermo Fisher | Cat#15596026 | |
| Chemical compound or drug | g-32P-ATP | PerkinElmer | Cat# BLU502H250UC | |
| Software, algorithm | ImageQuant TL | GE Healthcare/Life Sciences | RRID:SCR_014246 | Version 8.2, used for densitometry |
| Software, algorithm | PONDR | Molecular Kinetics, Inc, Washington State University and the WSU Research Foundation | pondr.com | Used to predict disordered regions in Csm5 |
| Software, algorithm | PyMOL | The PyMOL Molecular Graphics System, version 2.0 Schrödinger, LLC | RRID:SCR_000305 | Version 2.5, used for structural analyses |
| Software, algorithm | CRISPR-Cas10 Protospacer Selector Tool | ahatoum/CRISPR-Cas10-Protospacer-Selector is licensed under the GNU General Public License v3.0 (*Bari et al., 2017*) | https://github.com/ahatoum/CRISPR-Cas10-Protospacer-Selector | Used to predict protospacer sequence to target phage Andhra. |

## Bacterial strains, phages, and growth conditions

*S. aureus* RN4220 was propagated in Tryptic Soy Broth (TSB) medium (BD Diagnostics, NJ). *S. epidermidis* LM1680 and RP62a were propagated in Brain Heart Infusion (BHI) medium (BD Diagnostics). *Escherichia coli* DH5α was propagated in Luria-Bertani (LB) broth (VWR, PA), and *E. coli* BL21 (DE3) was propagated in Terrific broth (TB) medium (VWR) for protein purification. Corresponding media were supplemented with the following: 10 µg/ml chloramphenicol (to select for p*crispr-spc-*, p*crispr-cas-*, and pKOR1-based plasmids), 15 µg/ml neomycin (to select for *S. epidermidis* cells), 5 µg/ml mupirocin (to select for pG0400-based plasmids), 50 µg/ml kanamycin (to select for pET28b-His10Smt3-based plasmids), and 30 µg/ml chloramphenicol (to select for *E. coli* BL21 [DE3]). Phages CNPx and ISP were propagated using *S. epidermidis* LM1680 as host, and phage Andhra was propagated using *S. epidermidis* RP62a as host. For phage propagation, overnight cultures of the corresponding hosts were diluted at 1:100 in BHI supplemented with 5 mM $CaCl_2$ and phage ($10^6$–$10^8$ pfu/ml). Cultures were incubated at 37°C with agitation for 5 hr. One-fifth of the volume of host cells (grown to mid-log) was added into the bacteria-phage mixture and incubated for an additional 2 hr at 37°C with agitation. Cells were pelleted at 5000 × *g* for 5 min at 4°C, and the supernatant containing phage was filtered using a 0.45 µm syringe filter. Phage titers were determined using the double-agar overlay method as described in *Cater et al., 2017*. Briefly, a semisolid layer of 0.5× heart infusion agar (HIA) medium (Hardy Diagnostics, CA) containing 5 mM $CaCl_2$ and a 1:100 dilution of overnight host culture was overlaid atop a solid layer of Tryptic Soy Agar (TSA) (BD Diagnostics) plates supplemented with 5 mM $CaCl_2$. Filtered phages were diluted in tenfold dilutions and spotted atop the semisolid layer, spots were air-dried, and plates were incubated overnight at 37°C. Plaques were then enumerated, and phage titers in plaque-forming units/ml (pfu/ml) were determined.

## Construction of pKOR1-based plasmids and transformation into *S. epidermidis* LM1680

The pKOR1 system (*Bae and Schneewind, 2006*) was used to create in-frame deletions of *rnr* (encodes RNase R) and to reinsert an *rnr* variant (*rnr**, which has two silent mutations, see *Figure 1—figure supplement 1*) into *S. epidermidis* LM1680 WT and Δ*pnp* strains, and RP62a WT and Δ*pnp* strains. The pKOR1 vector was used as a template to amplify the backbone for all pKOR1-based constructs with primers A481/L138 via PCR amplification. The plasmid, pKOR1-Δ*rnr*, was created using a three-piece Gibson assembly (*Gibson et al., 2009*) and used to delete *rnr*. Briefly, two DNA fragments flanking upstream and downstream of *rnr* were obtained via PCR amplification using primers L139/L140 and L141/L142 (*Supplementary file 2*), respectively, and *S. epidermidis* RP62a WT as template. The PCR products of these flanks and pKOR1 backbone were purified using the EZNA Cycle Pure Kit (Omega Bio-tek, CA) and Gibson assembled. The plasmid pKOR1-*rnr** was created via a three-piece Gibson assembly and used to reintroduce *rnr** back to Δ*rnr* strains as follows. Briefly, primers L154/L155 (which bind to *rnr*) were designed to introduce two silent mutations that remove a native EcoR1 restriction site (*Figure 1—figure supplement 1C and D* and *Supplementary file 2*). Then, upstream and downstream flanking regions of *rnr* were amplified with PCR using primers L139/L155 and L154/L142, respectively, with *S. epidermidis* RP62a WT as template. PCR products of these flanks and pKOR1 backbone were purified using the EZNA Cycle Pure Kit and Gibson assembled. All assembled constructs were transformed via electroporation into *S. aureus* RN4220. Four transformants were selected for each construct and the presence of the plasmid was confirmed using PCR amplification and DNA sequencing with primers L145/L146 (*Supplementary file 2*). Confirmed plasmids were extracted using the EZNA Plasmid DNA Mini Kit (Omega Bio-tek) and introduced into *S. epidermidis* LM1680 WT and Δ*pnp* via electroporation. Plates were incubated at 30°C for 48 hr. Four transformants were selected and analyzed using PCR amplification and DNA sequencing with primers L145/L146 to confirm the presence of plasmid. Confirmed *S. epidermidis* LM1680 WT and Δ*pnp* transformants were used to proceed with mutagenesis and *S. epidermidis* LM1680 WT harboring appropriate plasmid was used to transfer the pKOR1-based plasmids into *S. epidermidis* RP62a WT and Δ*pnp* using phage-mediated transduction.

## Transduction of pKOR1-based plasmids into *S. epidermidis* RP62a

The temperate phage CNPx was used to transduce pKOR1-based plasmids from *S. epidermidis* LM1680 WT into *S. epidermidis* RP62a WT and Δ*pnp* as described previously in *Chou-Zheng and Hatoum-Aslan, 2019* with slight modifications. Briefly, overnight cultures of *S. epidermidis* LM1680 WT and Δ*pnp* strains harboring pKOR1-based plasmids were used to propagate phage CNPx as described above. Bacteria-phage cultures were incubated at 37°C with agitation for 5 hr, or until cell lysis. Cells were pelleted at 5000 × *g* for 5 min at 4°C, and the phage lysates were passed through a 0.45 µm syringe filter. Filtered phage lysates were then mixed with mid-log *S. epidermidis* RP62a cells in a 1:10 dilution and incubated at 37°C for 20 min. Bacteria-phage cultures were pelleted at 5000 × *g* for 1 min. Cell pellets were washed twice with 1 ml of fresh BHI, and the final pellets were resuspended in 200 µl of fresh BHI and plated entirety onto BHI agar containing appropriate antibiotics. Plates were then incubated at 30°C for 48 hr. Four transductants were selected for each construct and the plasmid's presence was confirmed using PCR amplification and DNA sequencing with primers L145/L146 (*Supplementary file 2*).

## Generation of *S. epidermidis* Δ*rnr* and Δ*rnr*Δ*pnp*

*S. epidermidis* strains bearing pKOR1-Δ*rnr* and pKOR1-*rnr** were used to generate all corresponding mutants using allelic replacement (*Bae and Schneewind, 2006*) as described previously in *Chou-Zheng and Hatoum-Aslan, 2019*. Four independent Δ*rnr* and Δ*rnr*Δ*pnp* deletion strains (i.e., biological replicates) were created and confirmed using PCR amplification and DNA sequencing with primers L143/L157. Four independent Δ*rnr*::*rnr** and Δ*rnr*Δ*pnp*::*rnr** complemented strains (i.e., biological replicates) were created and confirmed using three methods: PCR amplification, DNA sequencing with primers L143/L153, and EcoRI (New England Biolabs, MA) digestion of PCR products (*Supplementary file 2*).

## Construction of p*crispr-cas*/*csm5*^H6N Δ18, Δ31, and Δ46

All p*crispr-cas*-based plasmids were constructed using a three-piece Gibson assembly. The p*crispr-cas* plasmid (**Hatoum-Aslan et al., 2013**) was used as a template to amplify the backbone for these constructs. The three PCR products for p*crispr-cas*/*csm5*^H6NΔ18 were obtained using primer sets F063/ F066, F067/L247, and F062/L246 (**Supplementary file 2**). The three PCR products for p*crispr-cas*/ *csm5*^H6NΔ31 were obtained using primer sets F061/F066, F067/L265, and F046/L264. The three PCR products for p*crispr-cas*/*csm5*^H6NΔ46 were obtained using primer sets F061/F066, F067/L275, and F046/L274. All PCR products were purified using the EZNA Cycle Pure Kit and Gibson assembled. All assembled constructs were introduced into *S. aureus* RN4220 via electroporation. Four transformants were selected for each construct and confirmed to harbor the plasmid using PCR amplification and DNA sequencing with primers A416/F113. Confirmed constructs were extracted using the EZNA Plasmid DNA Mini Kit and transferred via electroporation into *S. epidermidis* LM1680 WT. Four transformants were selected and analyzed with PCR amplification and DNA sequencing using primers A416/F113 to confirm the presence of desired plasmids.

## Construction of *pcrispr-spc*-based plasmids

Spacers were designed using the protospacer selector tool (https://github.com/ahatoum/CRISPR-Cas10-Protospacer-Selector; **ahatoum, 2018**) described in **Bari et al., 2017**. Briefly, spacers were designed to target specific gene region of the corresponding phage, or the *nes* gene of conjugative plasmid pG0400, that bear no complementarity between the 8-nt tag on the 5′-end of the crRNA (5′-ACGAGAAC), and the 'anti-tag' region adjacent to the protospacer. Selected spacers were introduced into the template p*crispr-spcA1* (referred to as p*crispr-spcA2* in **Bari et al., 2017**) via inverse PCR using the primers listed in **Supplementary file 2**. All PCR products were purified using the EZNA Cycle Pure Kit. Linear products were phosphorylated with T4 Polynucleotide Kinase (New England Biolabs) for 1 hr at 37°C and circularized with T4 DNA Ligase (New England Biolabs) overnight at room temperature using buffers and instructions provided by the manufacturer. All assembled constructs were introduced into *S. aureus* RN4220 via electroporation. Four transformants were selected for each construct and confirmed via PCR amplification and DNA sequencing with primers A200/F052. Confirmed plasmids were extracted using the EZNA Plasmid DNA Mini Kit and introduced into *S. epidermidis* RP62a via electroporation. Four transformants were selected and analyzed with PCR amplification and DNA sequencing with primers A200/F052 to confirm presence of desired plasmids.

## CRISPR-Cas10 functional assays

Conjugation assays using pG0400-WT and pG0400-mut were carried out by filter mating *S. aureus* donor strains harboring these plasmids with various *S. epidermidis* recipient strains as described previously in **Walker and Hatoum-Aslan, 2017**. The conjugation data reported represents mean values (± SD) of 3–5 independent trials (see appropriate figure legends for details). Phage challenge assays were carried out by spotting tenfold dilutions of phages atop lawns of cells as previously described in **Chou-Zheng and Hatoum-Aslan, 2019**. Phage CNPx was used to infect *S. epidermidis* LM1680 WT and mutant strains carrying p*crispr-cas*-based plasmids. Phages Andhra and ISP were used to infect *S. epidermidis* RP62a WT and mutant strains carrying p*crispr-spc*-based plasmids. The phage challenge data reported represents mean values (± SD) of three independent trials.

## Purification of Cas10-Csm complexes from *S. epidermidis* LM1680

Cas10-Csm complexes containing a 6-His tag on the N-terminus of Csm2 or Csm5 in pcrispr-cas-based plasmids were overexpressed in *S. epidermidis* LM1680 cells, harvested, and stored exactly as described in **Chou-Zheng and Hatoum-Aslan, 2017**. Final pellets were purified following the first affinity chromatography protocol (Ni$^{2+}$-affinity chromatography) with slight modifications. Briefly, cell pellets were resuspended in 10 ml of lysis buffer A (22 mM MgCl$_2$, 44 μg/ml lysostaphin) and incubated in a water bath at 37°C for 1 hr. Lysed cells were then resuspended with 10 ml of lysis buffer B (50 mM NaH$_2$PO$_4$, 300 mM NaCl, pH 8.0) supplemented with 20 mM imidazole, 0.1% Triton X-100, and one cOmplete EDTA-free protease inhibitor tablet (Roche, Basel, Switzerland). Cells were homogenized by inverting the tube several times until the mixture becomes very viscous (a 5 min incubation period at room temperature might be necessary to achieve viscosity). Cells were then sonicated, and insoluble material was removed via centrifugation and filtration. Cleared lysates were passed through

a 5 ml gravity column (G-Biosciences, MO) containing 1.5 ml of Ni$^{2+}$-NTA agarose resin (Thermo Fisher Scientific, MA) pre-equilibrated with lysis buffer B. Nickel-bound complexes were then washed with 15 ml of lysis buffer B supplemented with 20 mM imidazole and 5% glycerol, followed by another 15 ml wash of lysis buffer B supplemented with 20 mM imidazole and 10% glycerol. Complexes were then eluted with five 600 µl aliquots each of lysis buffer B supplemented with 250 mM imidazole and 10% glycerol. Complexes were resolved on a 15% SDS-PAGE and visualized with Coomassie G-250. A pre-stained protein standard (New England Biolabs) was used to estimate molecular weight. Protein concentrations were determined using absorbance measurements at 280 nm (A280) with a Nano-Drop2000 spectrophotometer (Thermo Fisher Scientific).

## Reconstitution of crRNA maturation

300 pmol of Cas10-Csm complexes purified from LM1680/$\Delta rnr\Delta pnp$ were combined with 100 pmol of purified Cbf1, PNPase, and/or RNase R in Nuclease Buffer A (25 mM Tris-HCl pH 7.5, 2 mM DTT) supplemented with 10 mM MgCl$_2$ (PNPase and RNase R), or 10 mM MnCl$_2$ (Cbf1). Nuclease reactions were carried out at 37°C for 30 min, or for a time course of 15, 30, and 60 min. Reactions were halted on ice for 10 min, and then crRNAs were extracted and visualized.

## Extraction and visualization of crRNAs

Total crRNAs were extracted from purified Cas10-Csm complexes as described previously in *Chou-Zheng and Hatoum-Aslan, 2019* with slight modifications. Briefly, 300–600 pmols of purified complexes were resuspended in 750 µl TRIzol Reagent (Invitrogen, NY) and subsequent RNA extraction steps were completed as recommended by the manufacturer. Extracted crRNAs were end-labeled with T4 Polynucleotide Kinase in a reaction containing γ-[$^{32}$P]-ATP (PerkinElmer, MA), and resolved on an 8% Urea PAGE. The gel was exposed to a storage phosphor screen and visualized using an Amersham Typhoon biomolecular imager (Cytiva, MA). For densitometric analysis, the ImageQuant software was used. Percent of intermediate crRNAs was obtained using the following equation: [intensity of intermediate crRNA signal (71 nt) ÷ sum of signal intensities for the dominant crRNA species (71 nt +43 nt + 37 nt + 31 nt)]×100%. The data reported represents mean values (± SD) of 2–4 independent trials (see appropriate figure legends for details).

## Construction of pET28b-His$_{10}$Smt3-based plasmids

pET28b-His$_{10}$Smt3-*csm5Δ46* was constructed via inverse PCR using primers L274/L275 (*Supplementary file 2*) and template pET28b-His$_{10}$Smt3-*csm5* (*Walker et al., 2017*). PCR products were digested with DpnI (New England Biolabs) as indicated by the manufacturer and purified using the EZNA Cycle Pure Kit. Purified PCR products were then 5′-phosphorylated with T4 Polynucleotide Kinase for 1 hr at 37°C and circularized with T4 DNA Ligase overnight at room temperature using buffers and instructions provided by the manufacturer. Ligated pET28b-His$_{10}$Smt3-*csm5Δ46* constructs were introduced into *E. coli* DH5α via chemical transformation. Three transformants were selected, screened, and confirmed using PCR amplification and DNA sequencing with primers T7P/T7T. Confirmed plasmids were purified using the EZNA Plasmid DNA Mini Kit and introduced into *E. coli* BL21 (DE3) via chemical transformation for protein purification. Three transformants were selected and reconfirmed using PCR amplification and DNA sequencing with primers T7P/T7T (*Supplementary file 2*).

## Overexpression and purification of recombinant Csm5, Csm5Δ46, PNPase, RNase R, and Cbf1 from *E. coli*

*E. coli* BL21 (DE3) cells bearing pET28b-His$_{10}$Smt3-based plasmids were grown, induced, and recombinant proteins purified as previously described (*Walker et al., 2017*) with slight modifications. Following cell harvesting, pellets were placed on ice and resuspended in 30 ml of Buffer A (50 mM Tris-HCl pH 6.8, 1.25 M NaCl, 200 mM Li$_2$SO4, 10% sucrose, 25 mM imidazole) supplemented with one cOmplete EDTA-free protease inhibitor tablet (Roche), 0.1 mg/ml lysozyme, and 0.1% Triton X-100. Cells were incubated for 1 hr at 4°C with constant rotation, then sonicated. Insoluble material was removed via centrifugation and filtration. Cleared lysates were mixed with 4 ml of Ni$^{2+}$-NTA agarose resin pre-equilibrated with Buffer A, then mixed for 1 hr at 4°C with constant rotation. The resin was pelleted, washed with 40 ml of Buffer A, and pelleted again. The resin was then resuspended in 5 ml of Buffer A and transferred to a 5 ml gravity column. The resin was further washed with 20 ml

of Buffer A. Proteins were eluted stepwise with three aliquots of 1 ml each of IMAC buffer (50 mM Tris-HCl pH 6.8, 250 mM NaCl, 10% glycerol) containing 50, 100, 200, and 500 mM imidazole. Eluted protein fractions were resolved on a 15% SDS-PAGE, visualized with Coomassie G-250, and estimated molecular weight was determined with pre-stained protein standard. Fractions containing the desired protein were pooled and mixed with SUMO Protease (MCLAB, CA) with the provided SUMO buffer (salt-free). The mixtures were dialyzed for 3 hr against IMAC buffer containing 25 mM imidazole. The dialysate was mixed with 2 ml of $Ni^{2+}$-NTA agarose resin (pre-equilibrated with IMAC buffer containing 25 mM imidazole) and mixed for 1 hr with constant rotation at 4°C. The digested mixture was passed through a 5 ml gravity column, and the untagged protein was collected in the flow-through. Additional untagged protein was collected by flowing through the column two 1 ml aliquots each of IMAC buffer containing 50, 100, and 500 mM imidazole. Proteins were resolved, visualized, and estimated as described above. Protein concentrations less than 1 mg/ml were concentrated using a 10K MWCO centrifugal filter (Pall Corporation, NY). Protein concentrations were determined using absorbance measurements at 280 nm (A280) with a NanoDrop2000 spectrophotometer.

## Nuclease assays

A 31-nt single-stranded RNA substrate (5′ ACGAGAACACGUAUGCCGAAGUAUAUAAAUC) was 5′ end-labeled with T4 Polynucleotide Kinase and γ-[32P]-ATP, and purified with a G25 column (IBI Scientific, IA). Labeled substrate was incubated with 1 pmol of PNPase and 5 pmol of Csm5 WT or Csm5Δ46 in Nuclease Buffer B (25 mM Tris-HCl pH 7.5, 2 mM DTT, 10 mM $MgCl_2$). Nuclease reactions were carried out at 37°C in a time course of 0.5, 5, and 15 min, and quenched by adding an equal volume of 95% formamide loading buffer. Reactions were resolved on a 15% UREA PAGE. Gel was exposed to a storage phosphor screen and visualized using an Amersham Typhoon biomolecular imager.

## Native gel electrophoresis

All native electrophoresis gels were run in a Tetra Vertical Electrophoresis Chamber (Mini-PROTEAN Tetra Cell, Bio-Rad, CA). Recombinant PNPase (100 pmol) alone or in combination with Csm5 WT, or Csm5Δ46, (25, 50, and 100 pmol) was resolved in a 6% native polyacrylamide gel (29:1 acrylamide/bisacrylamide) of 0.75 mm thick. Tris-glycine buffer (25 mM Tris, 250 mM glycine, pH 8.5) was used to prepare and run the gels. Recombinant RNase R (30 pmol) or BSA (Sigma-Aldrich, MO) were alone or combined with Csm5 WT or Csm5Δ46 (90, 180, 225, and 270 pmol) were resolved in 6% native polyacrylamide gels (29:1 acrylamide/bisacrylamide) of 1.0 mm thick. Tris-CAPS buffer (60 mM Tris, 40 mM CAPS, pH 9.3–9.6) was used to prepare and run the gels. Native gel electrophoresis was conducted on an ice-water bath for 100–130 min at 90 V. Proteins were visualized with Coomassie G-250. Native-Mark Protein Standard (Thermo Fisher Scientific) was used to estimate molecular weight.

## Affinity pulldown assays

Csm5-His10-Smt3 (6 nmol) was loaded onto columns containing 300 µl of $Ni^{2+}$-NTA-agarose resin that was pre-equilibrated with protein buffer (IMAC buffer containing 25 mM imidazole). Columns containing Csm5-His10-Smt3 were then washed with 2 ml of protein buffer to remove unbound protein. Next, RNase R (1 nmol) or protein buffer was passed through columns pre-loaded with Csm5-His10-Smt3. Columns were washed twice with 2 ml of protein buffer to remove unbound proteins. Proteins bound to the column were then eluted with IMAC buffer containing 500 mM imidazole in three separate elutions (300 µl, 400 µl, and 400 µl, respectively). As a negative control, RNase R was added to a pre-equilibrated column without Csm5-His10-Smt3. Washes and elutions collected at each step were resolved on denaturing SDS-PAGE gels and visualized with Coomassie G-250. The assay was repeated three independent times.

## Statistical analyses and replicate definitions

All graphed data represent the mean (± SD) of $n$ replicates, where the $n$ value is indicated in figure legends and source data files. Average values were analyzed in pairwise comparisons using two-tailed $t$-tests, and p-values $< 0.05$ were considered statistically significant. Sample sizes were empirically determined, and no outliers were observed or omitted. The following terms are used to describe the types of repetitions where appropriate in figure legends and source data files: independent trials, independent transformants, and biological replicates. Independent trials refer to repetitions of the

same experiment conducted at different times; independent transformants refer to one bacterial strain into which a construct was independently introduced; and biological replicates refer to different bacterial mutants that were independently created.

## Materials availability

All bacterial strains and constructs generated in this study can be made available by the corresponding author (AH-A) upon written request.

## Code availability

There was no new code generated in this work; however, a previously generated code was reused to identify permissive protospacers in the *nes* gene for type III-A CRISPR-Cas immunity (in *Figure 6D and E*). For these experiments, spacers were designed using the publicly available protospacer selector tool (https://github.com/ahatoum/CRISPR-Cas10-Protospacer-Selector) described in *Bari et al., 2017*.

## Acknowledgements

We acknowledge funding for this project from the National Science Foundation CAREER award (MCB/2054755). AH-A is also funded by an Investigators in the Pathogenesis of Infectious Disease award from the Burroughs Wellcome Fund.

## Additional information

### Funding

| Funder | Grant reference number | Author |
|---|---|---|
| National Science Foundation | MCB/2054755 | Asma Hatoum-Aslan |
| Burroughs Wellcome Fund | 1020298 | Asma Hatoum-Aslan |

The funders had no role in study design, data collection and interpretation, or the decision to submit the work for publication.

### Author contributions

Lucy Chou-Zheng, Data curation, Formal analysis, Investigation, Methodology, Writing - review and editing; Asma Hatoum-Aslan, Conceptualization, Data curation, Formal analysis, Supervision, Funding acquisition, Investigation, Visualization, Writing - original draft, Writing - review and editing

### Author ORCIDs

Asma Hatoum-Aslan (iD) http://orcid.org/0000-0003-2395-8900

### Decision letter and Author response

Decision letter https://doi.org/10.7554/eLife.81897.sa1
Author response https://doi.org/10.7554/eLife.81897.sa2

## Additional files

### Supplementary files

• Supplementary file 1. Accompanies *Figures 2, 3 and 5*, *Figure 3—figure supplement 1*, and *Figure 5—figure supplement 1*. Theoretical molecular weights and isoelectric points of purified proteins in this study.

• Supplementary file 2. DNA oligonucleotides used for cloning and PCR in this study.

• MDAR checklist

### Data availability

All data generated or analyzed during this study are included in the manuscript and supporting files. Source data files have been provided for Figures 1, 2, 3, 4, 5, 6, Figure 1-figure supplement 1, Figure

2-figure supplement 1, Figure 3-figure supplement 1, Figure 3-figure supplement 2, Figure 5-figure supplement 1, and Figure 6-figure supplement 1. Corresponding source data files are called: Figure 1-source data 1, Figure 1-source data 2, Figure 1-source data 3, Figure 2-source data 1, Figure 2-source data 2, Figure 2-source data 3, Figure 2-source data 4, Figure 3-source data 1, Figure 3-source data 2, Figure 4-source data 1, Figure 4-source data 2, Figure 4-source data 3, Figure 5-source data 1, Figure 5-source data 2, Figure 5-source data 3, Figure 6-source data 1, Figure 6-source data 2, Figure 1-figure supplement 1-source data 1, Figure 1-figure supplement 1-source data 2, Figure 2-figure supplement 1-source data 1, Figure 3-figure supplement 1-source data 1, Figure 3-figure supplement 2-source data 1, Figure 3-figure supplement 2-source data 2, Figure 3-figure supplement 2-source data 3, Figure 5-figure supplement 1-source data 1, Figure 6-figure supplement 1-source data 1, Figure 6-figure supplement 1-source data 2, and Figure 6-figure supplement 1-source data 3.

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
