## [Editor Report]

CRISPR-Cas systems are essential components of an adaptive immune system that protects bacteria and archaea from infection of foreign genetic elements like phages and plasmids. The work presented here demonstrates that some CRISPR systems (i.e., type III-A) rely on host nucleases (i.e., RNase R and PNPase) for faithful processing of CRISPR RNAs into short mature CRISPR RNA (crRNAs) that are required for defense. Collectively, this work expands our fundamental understanding of degradosome-associated nucleases, and their contribution to the adaptive immune response in bacteria.

---

## [Decision Letter]

**Decision letter after peer review:**

Thank you for submitting your article "Critical roles for 'housekeeping' nucleases in Type III CRISPR-Cas immunity" for consideration by *eLife*. Your article has been reviewed by 2 peer reviewers, one of whom is a member of our Board of Reviewing Editors, and the evaluation has been overseen by Bavesh Kana as the Senior Editor. The reviewers have opted to remain anonymous.

Essential revisions:

1. Based on results in Figure 1E, the authors claim "RNase R alone causes complete loss of precisely-processed mature species and production of crRNAs with a range of aberrant lengths, as well as moderate accumulation of 71 nt intermediates." However, the "moderate accumulation of 71 nt intermediates" is very weak, not quantified, and is not supported by the complementation assay of rnr, which has more 71 nt intermediates than either the WT or the deletion mutant. Quantify data and modify claims for accuracy.

2. Add the Ni-affinity purified Cas10-Csm complex from LM1680/(△pnp/△rnr) cells on the gel presented in Figure 2B.

3. The authors rely on gel shifts to show a physical association between RNase R and Csm5. However, Csm5 doesn't enter the gel for reasons explained by the π and RNase R oligomerizes in a concentration-dependent manner. These factors complicate the interpretation of the gel shift. The authors should complement the gel shift with an alternative method. One referee suggests adding a tag to Csm5 that can be detected by Western blot the other suggests isothermal titration calorimetry (ITC). Acceptance for publication does not require a physical interaction but testing this interaction using an alternative method is required.

4. The authors used the structure of S. thermophilus Csm5 to guide their design of truncations to probe potential intrinsically disordered regions (IDR1 and IDR2) that may be sites of interaction with PNPase or RNase R. Since the authors submitted their manuscript, an AlphaFold predicted structure of the S. epidermidis Csm5 has been released on the AlphaFold Protein Structure Database. In this model, the IDR2 region is predicted by AlphaFold to be a β strand at the center of a β sheet, rather than a disordered region. If the prediction is accurate, deletion of this strand could cause Csm5 to misfold, making it difficult to interpret what causes loss of interaction with PNPase (i.e. deletion of a specific interaction surface versus misfolding of the overall tertiary structure). In light of this, the discussion surrounding these experiments should be altered to include more caveats about the truncations, and conclusions based on this experiment should be softened.

*Reviewer #1 (Recommendations for the authors):*

1. It is possible with the AlphaFold structural model of S. epidermidis Csm5 (or using AlphaFold Multimer) that the authors may be able to design more conservative truncations or point mutations that block the interaction between PNPase and Csm5. While AlphaFold only provides a structural prediction and should be taken with a grain of salt, this particular sequence is predicted with high confidence, which has generally been found to provide highly accurate structural models. This may therefore be a better alternative to mapping potential interaction regions than comparing to an ortholog that could have gaps/insertions in comparison to the sequence of interest.

2. For the native gels testing an interaction between Csm5 and RNase R, have the authors tested whether Csm5 is present in the shifted band (e.g. by adding a tag to Csm5 and performing a Western blot)? This would substantially strengthen the conclusions that could be drawn from this experiment.

3. In the Discussion section, the authors compare PNPase and RNase R requirements to that of Csm6, stating that all three appear to be dispensable for phage defense when targeting high-abundance transcripts. Have the authors tested anti-phage activity upon deletion of RNase R and PNPase in a strain lacking Csm6? If the three nucleases act synergistically, it is possible that deletion of all three may reduce the anti-phage activity that is not affected in strains lacking csm6 or pnp/rnr. Although I understand this is a lot to ask using the strain with an endogenous CRISPR-Cas locus, this experiment could potentially be done using the Lm1680 strains bearing pcrispr-cas in which csm6 is deleted.

*Reviewer #2 (Recommendations for the authors):*

Based on results in Figure 1E, the authors claim "RNase R alone causes complete loss of precisely-processed mature species and production of crRNAs with a range of aberrant lengths, as well as moderate accumulation of 71 nt intermediates." However, the "moderate accumulation of 71 nt intermediates" is very weak, not quantified, and is not supported by the complementation assay of rnr, which has more 71 nt intermediates than either the WT or the deletion mutant. Consider revising these statements for accuracy. The addition of deep sequencing would help clarify the identity of these intermediate RNA species and explain the role of RNase R in moving these intermediate RNAs into the mature fraction.

The authors should consider adding deletions of pnp alone to Figure 1, so readers can make a direct comparison between rnr and rnr/pnp double mutants. A pnp complementation (△pnp:: pnp*), and pnp/rnr complementation (△pnp/△rnr:: pnp*/rnr*) should also be added for the same reasons (i.e., side-by-side comparisons).

The authors should consider running the Ni-affinity purified Cas10-Csm complex from LM1680/(△pnp/△rnr) cells on the gel presented in Figure 2B.

The authors rely on gel shift to show a physical association between RNase R and Csm5. However, Csm5 doesn't enter the gel for reasons explained by the π and RNase R oligomerizes in a concentration dependent manner. These factors complicate the interpretation of the gel shift. The authors should consider complementing the gel shift with an alternative method (i.e., ITC) that would enable a quantitative measure of the binding affinity.

The authors speculate "that both nucleases may be recruited by the same/overlapping binding site(s) on Csm5, with one or the other allowed to occupy the site at any given time. Such transient and dynamic interactions are known to occur with proteins." In figure 2 the authors test either or both nucleases, but it may work testing one, then the other, to determine if the order of addition impacts RNA length or efficiency of processing.

"The results showed that while Csm5△46 maintains its interaction with RNase R (Figure 5—figure supplement 1)," The migration of the proteins is modest and complicated by the high PI. These experiments would benefit from another technique that provides a more quantifiable metric.

"striking stimulation of PNPase's nucleolytic activity occurs (Figure 5C)." I do not disagree that there is some stimulation, but it may not be as strike to some readers as it is to these authors. I recommend quantifying these data.

---

## [Author Response]

Essential revisions:1. Based on results in Figure 1E, the authors claim "RNase R alone causes complete loss of precisely-processed mature species and production of crRNAs with a range of aberrant lengths, as well as moderate accumulation of 71 nt intermediates." However, the "moderate accumulation of 71 nt intermediates" is very weak, not quantified, and is not supported by the complementation assay of rnr, which has more 71 nt intermediates than either the WT or the deletion mutant. Quantify data and modify claims for accuracy.

The fractions of intermediate crRNAs are indeed quantified and shown in Figure 1F. As described in the figure legend, the fraction of crRNA intermediates were quantified by dividing the density of the band comprising the 71 nt intermediate by the sum of densities of bands comprising the intermediate plus each processed mature species (43, 37, and 31 nt bands). Since the deletion of RNase R causes near complete loss of the precisely-processed mature species, the denominator in the ratio is much reduced, resulting in the apparent accumulation of intermediates.

We agree with the reviewers that “accumulation of the 71 nt intermediates” is probably not the best way to describe the phenotype in the RNase R deletion mutant and likely detracts from the more obvious defect in the mutant (i.e. the production of crRNAs with a range of aberrant lengths). Thus, we have removed “moderate accumulation of 71 nt intermediates” from the revised manuscript as recommended by the reviewers. Also, we opted to keep the quantification in Figure 1F as is because we believe it is valuable to show the more prominent differences in 71 nt intermediates in the other mutant strains.

2. Add the Ni-affinity purified Cas10-Csm complex from LM1680/(△pnp/△rnr) cells on the gel presented in Figure 2B.

This has been added as Figure 2B and the legend and text have been adjusted accordingly in the revised manuscript.

3. The authors rely on gel shifts to show a physical association between RNase R and Csm5. However, Csm5 doesn't enter the gel for reasons explained by the π and RNase R oligomerizes in a concentration-dependent manner. These factors complicate the interpretation of the gel shift. The authors should complement the gel shift with an alternative method. One referee suggests adding a tag to Csm5 that can be detected by Western blot the other suggests isothermal titration calorimetry (ITC). Acceptance for publication does not require a physical interaction but testing this interaction using an alternative method is required.

Thank you for this suggestion. We now have added an affinity pulldown assay as an alternative method to demonstrate the interaction between Csm5 and RNase R (Figure 3—figure supplement 2). In this assay, Csm5-His10-Smt3 is loaded onto a Ni^2+^-agarose column, the column is washed to remove unbound protein, and then untagged RNase R (or protein buffer) is allowed to flow through the column (Figure 3—figure supplement 2 A). Following extensive washing of unbound proteins, those remaining in the column are eluted three times using a buffer containing imidazole. Consistent with the weak/transient interaction observed between the two proteins, non-stoichiometric amounts of RNase R were found to co-elute with Csm5 (Figure 3—figure supplement 2 B and C). Importantly, untagged RNase R alone fails to stick to the column when subjected to the same wash and elution steps (Figure 3—figure supplement 2 D). This new data has been added to the revised manuscript and described in the narrative (lines 196-205 in the marked-up revised manuscript).

4. The authors used the structure of S. thermophilus Csm5 to guide their design of truncations to probe potential intrinsically disordered regions (IDR1 and IDR2) that may be sites of interaction with PNPase or RNase R. Since the authors submitted their manuscript, an AlphaFold predicted structure of the S. epidermidis Csm5 has been released on the AlphaFold Protein Structure Database. In this model, the IDR2 region is predicted by AlphaFold to be a β strand at the center of a β sheet, rather than a disordered region. If the prediction is accurate, deletion of this strand could cause Csm5 to misfold, making it difficult to interpret what causes loss of interaction with PNPase (i.e. deletion of a specific interaction surface versus misfolding of the overall tertiary structure). In light of this, the discussion surrounding these experiments should be altered to include more caveats about the truncations, and conclusions based on this experiment should be softened.

While this manuscript was under review, several cryo-EM structures of the Cas10-Csm complex from *S. epidermidis* were solved and reported (Smith *et al.,* 2022, *Structure*). In the unbound complex (PDB ID 7V02), IDR2 of Csm5 does indeed overlap with a short β strand, but it is flanked by loops/unstructured regions. In addition, of the 46 residues that we deleted in the Csm5D46 mutant, 20 residues are unresolved in the experimentally-determined structure, supporting the notion that this region is generally flexible. Also, it is unlikely that this and the other Csm5 deletion mutants are misfolded because they all retain the ability to associate with the complex (Figure 4B), and we were able to readily purify the mutant with the largest deletion (Csm5△46) without any issues (Figure 5).

To address this concern, we added panel D in Figure 4—figure supplement 1, which highlights the IDR regions in Csm5 from the recently-published *S. epidermidis* Cas10-Csm complex structure and integrated the observations mentioned above in the narrative (lines 241-247 in the marked-up revised manuscript). We also softened the conclusions based on these experiments (lines 276-278 in the marked-up revised manuscript):

“Taken together, these results suggest that the IDR2 region of Csm5 likely plays a role in the recruitment and stimulation of PNPase, while the binding site for RNase R may reside elsewhere in Csm5”.

Reviewer #1 (Recommendations for the authors):1. It is possible with the AlphaFold structural model of S. epidermidis Csm5 (or using AlphaFold Multimer) that the authors may be able to design more conservative truncations or point mutations that block the interaction between PNPase and Csm5. While AlphaFold only provides a structural prediction and should be taken with a grain of salt, this particular sequence is predicted with high confidence, which has generally been found to provide highly accurate structural models. This may therefore be a better alternative to mapping potential interaction regions than comparing to an ortholog that could have gaps/insertions in comparison to the sequence of interest.

This has been addressed under the essential revisions point number 4 above. We agree that the actual structure of Csm5 now allows us to design more conservative truncations, and this will certainly be explored in a follow-up study.

2. For the native gels testing an interaction between Csm5 and RNase R, have the authors tested whether Csm5 is present in the shifted band (e.g. by adding a tag to Csm5 and performing a Western blot)? This would substantially strengthen the conclusions that could be drawn from this experiment.

We haven’t yet tested for the presence of Csm5 using western blot, but plan on pursuing this avenue in the future. To bolster the claim of a direct interaction between Csm5 and RNase R, we performed an additional affinity pulldown assay (Figure 3—figure supplement 2). Details can be found under the essential revisions point number 3 above.

3. In the Discussion section, the authors compare PNPase and RNase R requirements to that of Csm6, stating that all three appear to be dispensable for phage defense when targeting high-abundance transcripts. Have the authors tested anti-phage activity upon deletion of RNase R and PNPase in a strain lacking Csm6? If the three nucleases act synergistically, it is possible that deletion of all three may reduce the anti-phage activity that is not affected in strains lacking csm6 or pnp/rnr. Although I understand this is a lot to ask using the strain with an endogenous CRISPR-Cas locus, this experiment could potentially be done using the Lm1680 strains bearing pcrispr-cas in which csm6 is deleted.

Thank you for the suggestion. This would be interesting to test in a follow-up study.

Reviewer #2 (Recommendations for the authors):Based on results in Figure 1E, the authors claim "RNase R alone causes complete loss of precisely-processed mature species and production of crRNAs with a range of aberrant lengths, as well as moderate accumulation of 71 nt intermediates." However, the "moderate accumulation of 71 nt intermediates" is very weak, not quantified, and is not supported by the complementation assay of rnr, which has more 71 nt intermediates than either the WT or the deletion mutant. Consider revising these statements for accuracy.

This has been addressed under the essential revisions point number 1 above.

The addition of deep sequencing would help clarify the identity of these intermediate RNA species and explain the role of RNase R in moving these intermediate RNAs into the mature fraction.

Thank you for this suggestion. We feel that the genetic and biochemical data presented provide strong enough support for the main conclusion in this manuscript that RNase R is involved in crRNA maturation. However, we agree that RNA-seq could be informative to determine if RNase R is needed to process a specific subset of crRNAs with common 3’-end sequences/structures. This seems appropriate for a follow-up study.

The authors should consider adding deletions of pnp alone to Figure 1, so readers can make a direct comparison between rnr and rnr/pnp double mutants. A pnp complementation (△pnp:: pnp*), and pnp/rnr complementation (△pnp/△rnr:: pnp*/rnr*) should also be added for the same reasons (i.e., side-by-side comparisons).

We intentionally excluded these data because most are already presented in the original paper (Chou-Zheng and Hatoum-Aslan, *eLife*, 2019) and we did not wish to show redundant information.

The authors should consider running the Ni-affinity purified Cas10-Csm complex from LM1680/(△pnp/△rnr) cells on the gel presented in Figure 2B.

This has been added as a separate panel, Figure 2B in the revised manuscript.

The authors rely on gel shift to show a physical association between RNase R and Csm5. However, Csm5 doesn't enter the gel for reasons explained by the π and RNase R oligomerizes in a concentration dependent manner. These factors complicate the interpretation of the gel shift. The authors should consider complementing the gel shift with an alternative method (i.e., ITC) that would enable a quantitative measure of the binding affinity.

This has been addressed under the essential revisions point number 4 above.

The authors speculate "that both nucleases may be recruited by the same/overlapping binding site(s) on Csm5, with one or the other allowed to occupy the site at any given time. Such transient and dynamic interactions are known to occur with proteins." In figure 2 the authors test either or both nucleases, but it may work testing one, then the other, to determine if the order of addition impacts RNA length or efficiency of processing.

That would be an interesting thing to try, perhaps in a more detailed follow-up study.

"The results showed that while Csm5△46 maintains its interaction with RNase R (Figure 5—figure supplement 1)," The migration of the proteins is modest and complicated by the high PI. These experiments would benefit from another technique that provides a more quantifiable metric.

Thanks again for this suggestion. As explained in the essential revisions point number 4 above, we used a pulldown assay to confirm Csm5 WT-RNase R weak interaction, which manifests as a subtle shift in a native gel. We attempted ITC, but this has proven very challenging with demonstrating Csm5 interactions (with PNPase or RNase R) because the Csm5 prep readily precipitates when kept at room temperature for an extended period. We are actively pursuing other more quantitative assays to measure the interactions between Csm5 and PNPase/RNase R and hope to have such data available in a follow-up manuscript.

"striking stimulation of PNPase's nucleolytic activity occurs (Figure 5C)." I do not disagree that there is some stimulation, but it may not be as strike to some readers as it is to these authors. I recommend quantifying these data.

We agree that this is not the most striking example of PNPase nuclease stimulation, and we have seen better in past iterations. It is difficult to quantify this data because the stimulation manifests as a shift towards smaller band lengths rather than a faster disappearance of uncut substrate. We have removed the word “striking” from the narrative to at least soften the description.